



# Seismic physics-based characterization of permafrost sites using surface waves

Hongwei Liu[1], Pooneh Maghoul[1,*], and Ahmed Shalaby[1]

[1]Department of Civil Engineering, University of Manitoba, 75A Chancellors Circle, Winnipeg, MB, Canada

**Correspondence:** Pooneh Maghoul (pooneh.maghoul@umanitoba.ca)

**Abstract.** The adverse effects of climate warming on the built environment in (sub)arctic regions are unprecedented and accelerating. Planning and design of climate-resilient northern infrastructure as well as predicting deterioration of permafrost from climate model simulations require characterizing permafrost sites accurately and efficiently. Here, we propose a novel algorithm for analysis of surface waves to quantitatively estimate the physical and mechanical properties of a permafrost site.

We show the existence of two types of Rayleigh waves (R1 and R2; R1 travels relatively faster than R2). The R2 wave velocity is highly sensitive to the physical properties (e.g., unfrozen water content, ice content, and porosity) of permafrost or soil layers while it is less sensitive to their mechanical properties (e.g., shear modulus and bulk modulus). The R1 wave velocity, on the other hand, depends strongly on the soil type and mechanical properties of permafrost or soil layers. In-situ surface wave measurements revealed the experimental dispersion relations of both types of Rayleigh waves from which relevant properties of

a permafrost site can be derived by means of our proposed hybrid inverse and multi-phase poromechanical approach. Our study demonstrates the potential of surface wave techniques coupled with our proposed data-processing algorithm to characterize a permafrost site more accurately. Our proposed technique can be used in early detection and warning systems to monitor infrastructure impacted by permafrost-related geohazards, and to detect the presence of layers vulnerable to permafrost carbon feedback and emission of greenhouse gases into the atmosphere.

## 1 Introduction

Permafrost is defined as the ground that remains at or below 0°C for at least two consecutive years. The upper layer of the ground in permafrost areas, termed as the active layer, may undergo seasonal thaw and freeze cycles. The thickness of the active layer depends on local geological and climate conditions such as vegetation, soil composition, air temperature, solar radiation and wind speed.

Within the permafrost, the distribution of ice formations is highly variable. Ground ice can be present under distinctive forms including (1) pore ice, (2) segregated ice, and (3) ice-wedge (Couture and Pollard, 2017; Mackay, 1972). Pore water, which fills or partially fills the pore space of the soil, freezes in-place when the temperature drops below the freezing point (Porter and Opel, 2020). On the other hand, segregated ice is formed when water migrates to the freezing front and it can cause excessive deformations in frost-susceptible soils. Frost-susceptible soils, e.g. silty or silty clay soils, have relatively high

capillary potential and moderate intrinsic permeability. During the winter months, ground ice expands as the ground freezes,



and forms cracks in the subsurface (Liljedahl et al., 2016). Ice wedges are large masses of ice formed over many centuries by repeated frost cracking and ice vein growth.

Design and construction of structures on permafrost normally follow one of two broad principles which are based on whether the frozen foundation soil in ice-rich permafrost is thaw-stable or thaw-unstable. This distinction is determined by the amount of ice content within the permafrost. Ice-rich permafrost contains ice in excess of its water content at saturation. The construction on thaw-unstable permafrost is challenging and requires remedial measures since upon thawing, permafrost will experience significant thaw-settlement and suffer loss of strength to values significantly lower than that for similar material in an unfrozen state. Consequently, remedial measures for excessive soil settlements or design of new infrastructure in permafrost zones affected by climate warming would require a reasonable estimation of the ice content within the permafrost (frozen soil). The rate of settlement relies on the mechanical properties of the foundation permafrost at the construction site. Furthermore, a warming climate can accelerate the microbial breakdown of organic carbon stored in permafrost and can increase the release of greenhouse gas emissions, which in return would accelerate climate change (Schuur et al., 2015).

Several in-situ techniques have been employed to characterize or monitor permafrost conditions. For example, techniques such as remote sensing (Bhuiyan et al., 2020; Witharana et al., 2020; Zhang et al., 2018), and ground penetrating radar (GPR) (Christiansen et al., 2016; Munroe et al., 2007; Williams et al., 2011) have been used to detect ice-wedge formations within the permafrost layers. Also, electrical resistivity tomography (ERT) has been extensively used to qualitatively detect pore-ice or segregated ice in permafrost based on the correlation between the electrical conductivity and the physical properties of permafrost (e.g., unfrozen water content and ice content) (Glazer et al., 2020; Hauck, 2013; Scapozza et al., 2011; You et al., 2013). The apparent resistivity measurement by ERT is higher in areas having high ice contents (You et al., 2013); however, at high resistivity gradients, the inversion results become less reliable, especially for the investigation of permafrost base (Hilbich et al., 2009; Marescot et al., 2003). Furthermore, in ERT investigations, the differentiation between ice and certain geomaterials can be highly uncertain due to their similar electrical resistivity properties (Kneisel et al., 2008). GPR has been also used for mapping the thickness of the active layer; however, its application is limited to a shallow penetration depth in conductive layers due to the signal attenuation and high electromagnetic noise in ice and water (Kneisel et al., 2008). It is worth mentioning that none of the above-mentioned methods characterizes the mechanical properties of permafrost layers.

Non-destructive seismic testing, including multi-channel analysis of surface waves (MASW) (Dou and Ajo-Franklin, 2014; Glazer et al., 2020), passive seismic test with ambient seismic noise (James et al., 2019; Overduin et al., 2015), seismic reflection (Brothers et al., 2016), and seismic refraction method (Wagner et al., 2019) have been previously employed to map the permafrost layer based on the measurement of shear wave velocity. In the current seismic testing practice, it is commonly considered that the permafrost layer (frozen soil) is associated with a higher shear wave velocity due to the presence of ice in comparison to unfrozen ground. However, the porosity and soil type can also significantly affect the shear wave velocity (Liu et al., 2020a). In other words, a relatively higher shear wave velocity could be associated to an unfrozen soil layer with a relatively lower porosity or stiffer solid skeletal frame, and not necessarily related to the presence of a frozen soil layer. Therefore, the detection of permafrost layer and permafrost base from only the shear wave velocity may lead to inaccurate and even misleading interpretations.





Here, we present a hybrid inverse and multi-phase poromechanical approach for in-situ characterization of permafrost sites using surface wave techniques. In our method, we quantify the physical properties such as ice content, unfrozen water content, and porosity as well as the mechanical properties such as the shear modulus and bulk modulus of permafrost or soil layers. Through the mechanical properties of the solid skeleton frame, we can also predict the soil type and the sensitivity of the permafrost layer to permafrost carbon feedback and emission of greenhouse gases to the atmosphere. We also determine the depth of the permafrost table and permafrost base. The role of two different types of Rayleigh waves in characterizing the permafrost is presented based on an MASW seismic investigation in a field located at SW Spitsbergen, Norway. Multiphase poromechanical dispersion relations are developed for the interpretation of the experimental seismic measurements at the surface based on the spectral element method. Our results demonstrate the potential of seismic surface wave testing accompanied with our proposed hybrid inverse and poromechanical dispersion model for the assessment and quantitative characterization of permafrost sites.

## 2 Methods

### 2.1 Methodology Overview

Figure 1 shows the overview of the proposed hybrid inverse and poromechanical approach for in-situ characterization of permafrost sites. We can obtain the experimental dispersion relations for R1 and R2 Rayleigh wave types from the surface wave measurements. Then, we use the experimental dispersion of R2 waves to characterize the physical properties of the layers. A random sample is initially generated to ensure that soil parameters are not affected by a local minimum. Then the forward three-phase poromechanical dispersion solver is used to compute the theoretical dispersion relation of the R2 wave. Therefore, we can rank samples based on the $L_2$ norm between the experimental and theoretical dispersion relations. Based on the ranking of each sample, the Voronoi polygons (Neighborhood sampling method) are used to generate better samples with a smaller objective function until the solution converges. We can select the best samples with the minimum loss function and obtain the most likely physical properties and thickness of the active layer, permafrost layer, and unfrozen ground. After obtaining the physical properties, the mechanical properties can be derived based on the dispersion relation of the R1 wave mode in a similar manner, as summarized in Figure 1h (optimization variables exclude the physical properties and the thickness of each layer in this process).

### 2.2 Rayleigh wave dispersion relations

We consider the frozen soil specimen to be composed of three phases: solid skeletal frame, pore-water, and pore-ice. Through the infinitesimal kinematic assumption (Equation C1), the stress-strain constitutive model (Carcione and Seriani, 2001) (Equation C2), and the conservation of momentum (Equation C3), the field equations can be written in the matrix form (Equation C4). The matrix $\bar{\rho}$, $\bar{b}$, $\bar{R}$ and $\bar{\mu}$ are given in D. The field equations can also be written in the frequency domain by performing



**Figure 1. (a)** A general schematic of the MASW test at a permafrost site **(b)** Dispersion relations of R1 and R2 waves obtained from the experimental measurements. **(c)** Initial guess of the physical properties of active layer, permafrost layer and unfrozen ground. **(d)** Calculation of the theoretical dispersion relation of R2 wave using the forward three-phase poromechanical dispersion solver. **(e)** Solution ranking based on $L_2$ norm for R2 dispersion relations (experimental vs theoretical) using the hybrid inverse and poromechanical approach. **(f)** Neighborhood sampling for the reduction of $L_2$ norm using the hybrid inverse and poromechanical approach. **(g)** Select the best samples based on the minimum $L_2$ norm and obtain the physical properties and thickness for each layer. **(h)** Repeat the steps for dispersion inversion (c-f) of R1 dispersion relation to derive the mechanical properties of active layer, permafrost layer and unfrozen ground. **(i)** Select the best samples based on the minimum $L_2$ norm and obtain the mechanical properties.



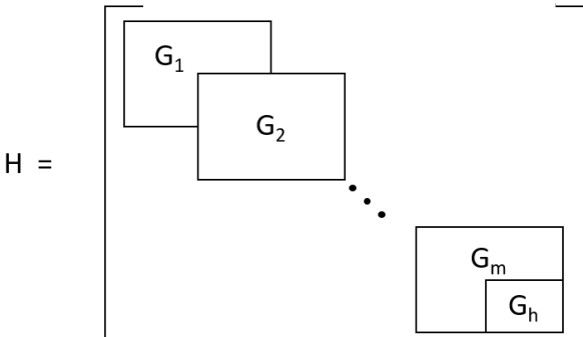

**Figure 2.** Construction of the global stiffness matrix

convolution with $e^{i\omega t}$. The field equations in the Laplace domain are obtained by replacing $\omega$ with $i \cdot s$ ($i^2 = -1$ and $s$ the Laplace variable).

To obtain the spectral element solution, the Helmholtz decomposition is used to decouple the P waves (P1, P2, and P3) and S waves (S1 and S2). The displacement vector ($\bar{u}$) is composed of the P wave scalar potentials $\phi$ and S wave vector potentials

$\bar{\psi} = (\psi_r, \psi_\theta, \psi_z)$. Since P waves exist in the solid skeleton, pore-ice and pore-water phases, three P wave potentials are used, including $\phi_s$, $\phi_i$ and $\phi_f$ (Equation C6). The detailed steps for obtaining the closed-form solutions for P waves and S waves using the Eigen decomposition are summarized in C. After obtaining the stiffness matrix for each layer, the global stiffness matrix, $H$, can be assembled by applying the continuity conditions at layer interfaces. The stiffness assembling method is shown in Figure 2.

The dispersion relation is obtained by setting a zero stress condition at the surface ($z = 0$). To obtain the non-trivial solution, the determinant of the global stiffness matrix has to be zero, as expressed in Equation 1 (Zomorodian and Hunaidi, 2006).

$$\det H(\omega, k) = 0. \tag{1}$$

The global stiffness matrix, $H(\omega, k)$, is a function of angular frequency $\omega$ and wavenumber $k$. For a constant frequency, the value of the wavenumber can be determined when the determinant of the global stiffness matrix is zero. The dispersion curve

is also commonly displayed as frequency versus phase velocity, $v = \frac{\omega}{k}$. The different wavenumbers determined at a given frequency correspond to dispersion curves of different modes. To extract the fundamental mode of the R1 wave, the velocities of P1 wave and S1 wave are calculated first for the given physical properties and mechanical properties of each layer. The global stiffness matrix for the R1 wave can be decomposed into the components related only to the P1 and S1 wave velocities. This is viable since we have proved that the R1 wave is generated by the interaction between the P1 and S1 waves. This approach

avoids the difficulties in differentiating the higher modes of R2 wave from the fundamental mode of the R1 wave. The detailed root search method has been documented in Liu et al. (2020b).





## 2.3 Inversion

The aim function is defined as the Euclidean norm between the experimental and numerical results of the dispersion relations.
The problem is formulated in Equation 2:

$$
\begin{cases}
minimize\ f(\boldsymbol{x}) = \frac{1}{2} \sum_{i=1}^{N} (y_i - \bar{y}_i(\boldsymbol{x}))^2 \\
subject\ to\ a_i \leq x_i \leq b_i,\ i = 1, \ldots, m
\end{cases}
\tag{2}
$$

where $f$ is the objective function; $\boldsymbol{x} = (x_1, x_2, ... x_m)$ is the optimization variable (e.g., porosity, and degree of saturation of unfrozen water, bulk modulus and shear modulus of solid skeleton frame as well as thickness of each layer); the constant $a_i$ and $b_i$ are limits or bounds for each variable; $m$ is the total number of variables; $y$ and $\bar{y}$ are the numerical and experimental
dispersion relations for the R1 or R2 waves.

Here, we used the neighborhood algorithm that benefits from the Voronoi cells to search the high-dimensional parameter space and reduce overall cost function (Sambridge, 1999). The algorithm contains only two tuning parameters. The neighborhood sampling algorithm includes the following steps: a random sample is initially generated to ensure the soil parameters are not affected by the local minima. Based on the ranking of each sample, the Voronoi polygons are used to generate better
samples with a smaller objective function. The optimization parameters are scaled between 0 and 1 to properly evaluate the Voronoi polygon limit. After generating a new sample, the distance calculation needs to be updated. Through enough iterations of these processes, the aim function can be reduced. The detailed description of the neighborhood algorithm is described by Sambridge (1999).

## 3   Identification of Rayleigh waves (R1 and R2) dispersion relations

From a poromechanical point of view, permafrost (frozen soil) is a multi-phase porous medium that is composed of a solid skeletal frame and pores filled with water and ice with different proportions. Here, we analyze the seismic wave propagation in permafrost based on the three-phase poroelastodynamic theory. Three types of P wave (P1, P2 and P3) and two types of S wave (S1, S2) coexist in three-phase frozen porous media (Carcione et al., 2000; Carcione and Seriani, 2001; Carcione et al., 2003). The P1 and S1 waves are strongly related to the longitudinal and transverse waves propagating in the solid skeletal frame,
respectively, but are also dependent on the interactions with pore ice and pore water (Carcione and Seriani, 2001). The P2 and S2 waves propagate mainly within pore ice (Leclaire et al., 1994). Similarly, the P3 wave is due to the interaction between the pore water and the solid skeletal frame. The velocity of different types of P waves and S waves is provided in A.

Here a uniform frozen soil layer is used to show the propagation of different types of P and S waves and subsequently the formation of Rayleigh waves (R1 and R2) at the surface. It is assumed that an impulse load with a dominant frequency of 100 Hz
is applied at the ground surface. The wave propagation analysis was performed in clayey soils by assuming a porosity (n) of 0.5, a degree of saturation of unfrozen water (Sr) of 50%, a bulk modulus (K) of 20.9 GPa and a shear modulus (G) of 6.85 GPa for the solid skeletal frame (Helgerud et al., 1999). The velocities of the P1 and P2 waves are calculated as 2,628 m/s and 910 m/s, respectively, based on the relations given in A. The velocity of P3 wave (16 m/s) is relatively insignificant in comparison to P1





and P2 wave velocities. Similarly, the velocities of the S1 and S2 waves are calculated as 1,217 m/s and 481 m/s, respectively.
Accordingly, the observed displacements measured at the ground surface with an offset from the impulse load ranging from 0 to 120 m are illustrated in Figure 3a. Figure 3b and 3c illustrate the appearance of two types of Rayleigh waves (R1 and R2) in a three-phase permafrost subsurface at 70 ms and 100 ms, respectively. Our results convincingly demonstrate that R1 waves appear due to the interaction of P1 and S1 waves. The phase velocity of R1 waves is slightly slower than the phase velocity of S1 waves. Similarly, the phase velocity of R2 waves is also slightly slower than the phase velocity of S2 waves. Briefly, the
order of phase velocities of different waves propagating within the domain is as follows: P1>P2>S1>R1>S2>R2>P3. The seismic measurements shown in Figure 3a are indeed a combination of both R1 and R2 waves.

The phase velocities of R1 and R2 waves are a function of physical properties (e.g., degree of saturation of unfrozen water, degree of saturation of ice, and porosity) and mechanical properties of the solid skeletal frame (e.g., bulk modulus and shear modulus). Figure 3d illustrates the effect of shear modulus and bulk modulus of the solid skeletal frame on the phase velocity
of R1 and R2 waves. Similarly, Figure 3e illustrates the effect of porosity and degree of saturation of ice on the phase velocity of R1 and R2 waves. It can be seen that the phase velocity of the R1 wave is mostly sensitive to the shear modulus of the solid skeletal frame; it is also dependent on the bulk modulus, porosity, and degree of saturation of ice. On the other hand, the phase velocity of the R2 wave is almost independent of the mechanical properties of the solid skeletal frame (Figure 3d), while it is strongly affected by the porosity and degree of saturation of ice (Figure 3e).

Our results also show that an increase in the degree of saturation of ice leads to an increase in the phase velocity of both types of Rayleigh waves. An increase in porosity leads to an increase in the phase velocity of R2. However, an increase in porosity may lead to either a decrease or an increase in the phase velocity of R1 wave, depending on the level of the degree of saturation of ice. Hence, we use the phase velocity of R2 waves identified by processing the seismic surface wave measurements to characterize the physical properties (e.g., porosity, degree of saturation of ice or degree of saturation of unfrozen water) of
permafrost or soil layers.

## 4 Case study for characterization of a permafrost site using surface wave technique

The case study site is located at the Fuglebekken coastal area in SW Spitsbergen, Norway (77°00'30"N and 15°32'00"E). The study area has a a thick layer of unconsolidated sediments that are suitable for near-surface geophysical investigations (Glazer et al., 2020). The unconsolidated sedimentary rock contains a high proportion of pore spaces; consequently, they can
accumulate a large volume of pore-water or pore-ice. From meteorological records, the mean annual air temperature (MAAT) at the testing site was historically below the freezing point, but more recently and due to a trend of climate warming, the MAAT recorded in 2016 is approaching 0°C (Glazer et al., 2020). Glazer et al. (2020) performed both seismic surveys (MASW test) and electrical resistivity investigations at the site in October 2017 to study the evolution and formation of permafrost considering surface watercourses and marine terrace. The MASW test was performed by using the geophone receivers distributed at 2
175 m spacing. Figure 4a shows the test site. Figure 4b illustrates the collected original seismic measurements at distances between 0 m and 120 m (hereafter referred to Section 1). The R1 and R2 Rayleigh waves are identified by visual inspection to obtain





**Figure 3. (a)** Theoretical time-series measurements for R1 and R2 Rayleigh waves at the ground surface **(b)** Displacement contour at time 70 ms. **(c)** Displacement contour at time 100 ms with the labeled R1 and R2 Rayleigh waves. **(d)** Effect of shear modulus and bulk modulus of the solid skeletal frame on phase velocity of R1 and R2 waves. **(e)** Effect of degree of saturation of ice on the phase velocity of R1 and R2 waves.

the experimental dispersion relations (Figure 4c and 4d). The phase velocity of R1 wave increases with frequency from 15 Hz to 75 Hz. The phase velocity of R2 wave decreases with frequency in the span of 17.5 Hz to 40 Hz.



In our simulations, the permafrost site is modeled as a three-layered system, consisting of an active layer at the surface
followed by a permafrost layer on top of the unfrozen ground. The ERT results reported by Glazer et al. (2020) proved that
the active layer is almost completely unfrozen during the MASW testing performed in October. The degree of saturation of
unfrozen water is considered above 85% for the active layer in our study. The temperature of the permafrost layer remains
below or at 0°C year round, but the volumetric ice content of the test site is unknown. Therefore, in our simulation, the degree
of saturation of unfrozen water in the permafrost layer is considered to be between 1% and 85% to be conservative. The
185 unfrozen ground is believed to have a degree of saturation of unfrozen water of about 100% (fully saturated). The porosity of
all three layers is distributed between 0.1 and 0.7. We previously showed that the dispersion relation of the R2 wave is strongly
dependent on the physical properties (e.g., porosity and degree of saturation of unfrozen water). Hence, the R2 dispersion
relation (Figure 4d) is used first to determine the most probable distributions of porosity and degree of saturation of unfrozen
water with depth. The other physical properties such as degree of saturation of ice, volumetric water content and volumetric
ice content can also be obtained by knowing porosity and degree of saturation of unfrozen water.

The mechanical properties of the solid skeletal frame in each layer are then obtained using the R1 wave dispersion relation.
The mechanical properties can be then used to determine whether the permafrost site is ice-rich. In fact, the direct detection of
the thin ice lenses using low frequency seismic waves is highly impossible due to the mismatch between the thickness of the
ice segregation layers and the wavelength generated in seismic tests. However, the mechanical properties of permafrost reveal
the mineral composition of the soil and soil type, which is valuable in the classification of ice-rich permafrost or even detection
of whether the permafrost layer is prone to greenhouse gases carbon dioxide and methane emission to the atmosphere.

Figure 5a shows the probabilistic distribution of the degree of saturation of unfrozen water with depth in Station 1. Our
results show that the active layer has a thickness of about 4.2 m with a degree of saturation of unfrozen water of 88%. The
ground temperature can be estimated given the unfrozen water content and porosity by an empirical relation described by Liu
et al. (2019). This implies the average soil temperature in the active layer is about 0°C. The predicted permafrost layer has a
thickness of about 14.6 m with a nearly 8.8%-22% of degree of saturation of unfrozen pore water. Given the high ice-to-water
ratio, we therefore interpret the permafrost is currently in a stable frozen state. Figure 5b shows the degree of saturation of ice
with depth. The degree of saturation of ice in the permafrost layer ranges from 77% to 91%. Figure 5c illustrates the porosity
distribution with depth. The porosity ranges from 0.34 to 0.38 in the first layer (active layer), from 0.43 to 0.46 in the second
layer (permafrost) and from 0.31 to 0.36 in the third layer (unfrozen ground). Figure 5d and 5e show the predicted mechanical
properties of the solid skeletal frame (shear modulus and bulk modulus) in each layer. The predicted shear modulus and bulk
modulus for the solid skeletal frame in the permafrost layer are about 11 GPa and 10 GPa, which are in the range for silty-
clayey soils (Vanorio et al., 2003). Figure 5f and 5g show the comparison between the numerical and experimental dispersion
relations for R2 and R1 waves, respectively. The numerical predictions are sufficiently close to the experimental dispersion
curves for both R1 and R2 waves.

Figure 6 illustrates the inversion process of the surface wave measurements for the R2 wave by means of the Neighborhood
algorithm. Initially, 20 random samples were employed in the entire space (to avoid the local minimum problem). Voronoi
decomposition is used to generate representative sampling points about the best samples in the previous steps. Figure 6a shows





**Figure 4.** Surface wave measurement in Section 1 (from 0 m to 120 m). **(a)** Study area in Holocene, Fuglebekken, SW Spitsbergen. **(b)** Waveform data from the measurements at different offsets in horizontal distance. **(c)** Experimental dispersion image for R1 wave. **(d)** Experimental dispersion image for R2 wave

the entire set of sampling points in the subspace between the degree of saturation of unfrozen water and the thickness of the active layer. Most sampling points are concentrated at the location where the degree of saturation of unfrozen water is 88% and the thickness of the active layer is 4.1 m. Similarly, in the subspace of the degree of saturation of unfrozen water and the thickness of the permafrost layer, our results showed that the permafrost layer is most likely having a thickness of 13.6 m with a degree of saturation of unfrozen water of 15%. Figure 6c shows the updates of each parameter (thickness, degree of saturation of unfrozen water and porosity) with the number of run in our forward solver. Our results show that the Neighborhood algorithm





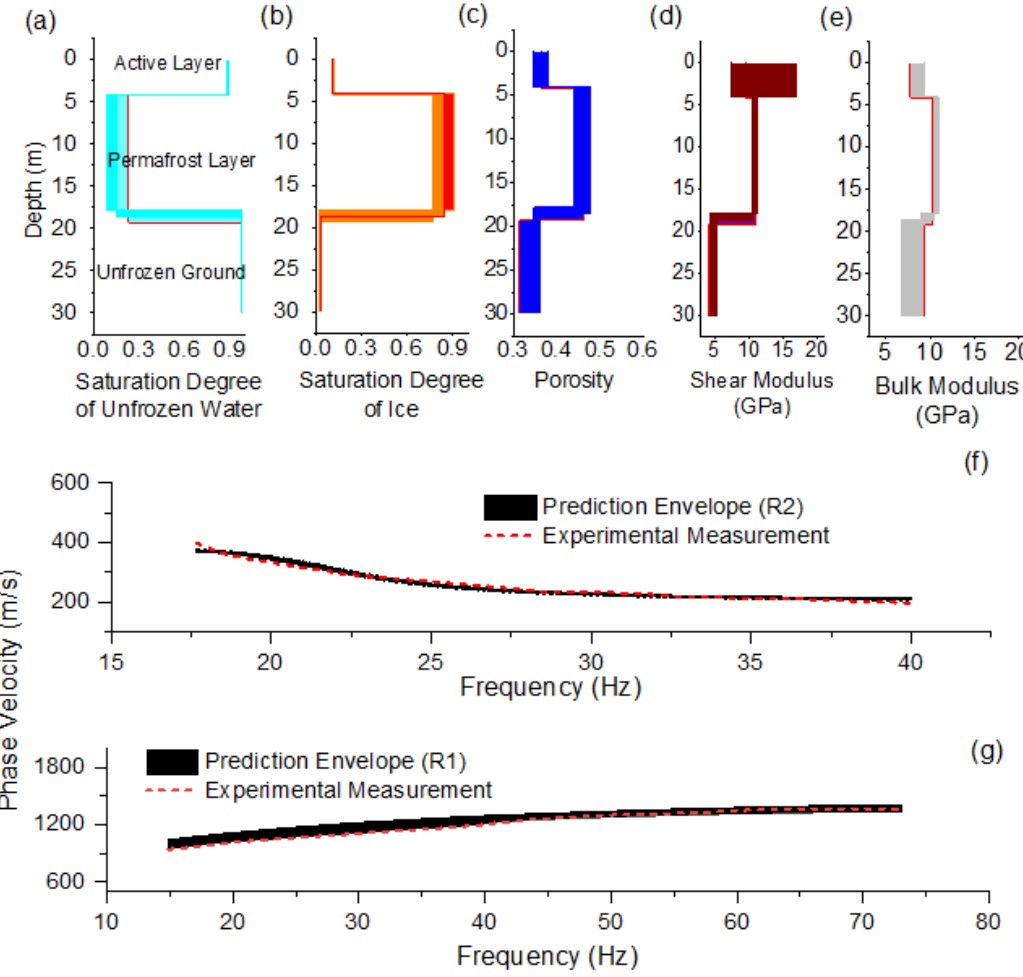

**Figure 5.** Surface wave inversion results for Section 1: 0m to 120m. **(a)** Degree of saturation of unfrozen water, **(b)** Degree of saturation of ice, **(c)** Porosity distribution, **(d)** Shear modulus of solid skeletal frame, **(e)** Bulk modulus of solid skeletal frame, **(f)** Experimental and numerical dispersion curves for R2 wave, **(g)** Experimental and numerical dispersion curves for R1 wave.

fully explores the searching space of each parameter. Figure 6c also illustrates that the solution converged after roughly 4,000 iterations and the loss function was reduced from 1000 to only 43 at the end.

We have previously shown the inversion process and results for Section 1 from 0 m to 120 m. Five additional sections spanning from 120 m to 600 m were also studied using a similar approach. The seismic measurements and dispersion relations for each section are given in B. Figure 7a shows the distribution of the degree of saturation of unfrozen water in the ground

based on the five independent MASW tests. The result demonstrates that the permafrost table is generally located at about 4 m below the ground surface, except at the offset distance from 360 m to 480 m where the permafrost table is located at 1.1 m below the ground surface. The thickness of the permafrost layer varies from 12 m to 15 m. We predicted that at the offset

**Figure 6.** Inversion process for the R2 wave dispersion relation. **(a)** Sampling subspace between the degree of saturation of unfrozen water and the thickness of the active layer. **(b)** Sampling subspace between the degree of saturation of unfrozen water and the thickness of the permafrost layer. **(c)** Updates of thicknesses of the active layer and permafrost layer as well as the physical properties in each layer by means of the Neighborhood algorithm

distance from 360 m to 480 m, the degree of saturation of ice is the highest (about 85%). Figure 7b illustrates the distribution of the predicted porosity in the test site. We also predicted a higher porosity of 0.66 at the offset distance from 360 m to 480 m.

The volumetric unfrozen water content (calculated as the product of porosity and the degree of saturation of unfrozen water) in the permafrost layer is about 0.08. Li et al. (2020); Zhang et al. (2020) showed that the residual volumetric unfrozen water content for silty-clay, clay, medium sand, and fine sand is 0.12, 0.08, 0.06 and 0.03, respectively. These predictions fit well within the reasonable range of volumetric unfrozen water content for clay or clayey silt. Sufficient agreement exists between the numerical and experimental dispersion relations for the R2 wave (Figure 7d) which confirms the acceptance of the predicted



**Figure 7.** Summary of the inversion results at the offset distance from 0 m to 600 m. **(a)** Volumetric ice content distribution. **(b)** Soil porosity distribution. **(c)** Distribution of the shear modulus of the solid skeletal frame. **(d)** Comparison between the numerical and experimental dispersion curves for R2 wave. **(e)** Predicted average soil temperature distribution.

values for the volumetric ice content (calculated as the product of porosity and the degree of saturation of ice) and porosity. Similarly, we obtain the mechanical properties of the solid skeletal frame for each layer (Figure 7c) based on the phase velocity of R1 waves. By an empirical relation between the unfrozen water content, porosity, and soil temperature (Liu et al., 2019), we compute the average ground temperature distribution in the test site. It is predicted that at the offset distance from 360 m to 480 m the coldest temperature of about -12 °C (Figure 7e) occurs in the permafrost layer, which is highly related to the high

ice content in this section.



## 5   Discussion and Conclusions

We developed a hybrid inverse and multi-phase poromechanical approach to quantitatively estimate the physical and mechanical properties of a permafrost site. The identification of two distinctive types of Rayleigh waves in the surface wave field measurements in permafrost sites is critical for quantitative characterization of the layers. The identification of the R2 wave

allows the quantitative characterization of physical properties of soil layers independently without making assumptions of the mechanical properties of the layers. This approach simplifies the inversion of the multi-layered three-phase poromechanical model since the dependent optimization variables are largely reduced. The inversion results from the R2 wave dispersion relation can be further used in the characterization of the mechanical properties of soil layers based on the R1 wave dispersion relation. This also increases the stability and convergence rate of the inversion solver and makes the analysis more efficient.

In ice-rich permafrost that contains ice in excess of the water content required to fill pore space in the unfrozen state (normally shown as ice lenses), the direct detection of the thin ice lenses using the surface waves is almost impossible due to the mismatch between the thickness of the ice segregation layers and the wavelength generated in the seismic tests. However, the mechanical properties of the solid skeletal frame can reveal the type of soil, which can be used to identify an ice-rich permafrost layer. Furthermore, the sensitivity of the permafrost layer to permafrost carbon feedback and emission of greenhouse gases (e.g.,

methane, carbon dioxide etc.) to the atmosphere can be determined. For example, if the mechanical properties of the solid skeletal frame correspond to the ones for peat we can perform more detailed investigation to assess the sensitivity of the permafrost to greenhouse gases emission.

We found that at the offset distance from 360 m to 480 m (Figure 7), the saturation degree of ice (85%) is significantly high and the value of the porosity is about 0.66, which could be due to the presence of ice segregation layers. Meanwhile, based on

the mechanical properties of the solid skeletal frame (most likely clayey or clayey silt soil), we can reasonably consider the permafrost layer at the offset distance from 360 m to 480 m to be ice-rich and ice segregation layers are expected to contribute to its relatively higher volumetric ice content. Here, we consider the segregated ice plays a same role as pore-ice from a continuum mechanics point of view. The growth of ice lenses is approximated as an increase in the soil porosity (Michalowski and Zhu, 2006). Therefore, the determined volumetric ice content (Figure 7) can correspond to both pore-ice and segregated

ice (ice lenses) as an average value.

Additional work on the characterization of permafrost should explore ways to reduce the uncertainty in the proposed hybrid inverse and multi-phase poromechanical approach. The uncertainty originates from the non-uniqueness in the inverse analysis (local minima problem) and the limited number of constraints in the inversion analysis. It is recommended to use other geophysical methods to improve the resolution and reduce uncertainty of the permafrost mapping. With the proposed seis-

mic wave-based method as the main investigation tool, ERT, GPR and electromagnetic (EM) Tomography can augment the investigation data and supply additional constraints to the inversion analysis.

The proposed hybrid inverse and multi-phase poromechanical approach can potentially be used for the design of an early warning system for permafrost by means of an active or passive seismic test. The seismic noise from traffic can generate stress waves as they travel on the permafrost foundation. Pre-installed geophones can be used to capture the propagation of R1 and





R2 waves. By applying the proposed signal processing approach, we can estimate the physical and mechanical properties of permafrost for monitored sites. The early warning system can provide long-term tracking of permafrost conditions particularly when the ice content or mechanical properties of permafrost approach critical values.

## Appendix A: Definition of Phase Velocities

The velocities of the three types of P waves are determined by a third degree characteristic equation:

$$\Lambda^3 \tilde{R} - \Lambda^2 \left( (\rho_{11}\tilde{R}_{iw} + \rho_{22}\tilde{R}_{si} + \rho_{33}\tilde{R}_{sw}) - 2(R_{11}R_{33}\rho_{23} + R_{33}R_{12}\rho_{12}) \right)$$

$$+ \Lambda \left( (R_{11}\tilde{\rho}_{iw} + R_{22}\tilde{\rho}_{si} + R_{33}\tilde{\rho}_{sw}) - 2(\rho_{11}\rho_{23}R_{23} + \rho_{33}\rho_{12}R_{12}) \right) - \tilde{\rho} = 0$$

where

$$\tilde{R} = R_{11}R_{22}R_{33} - R_{23}^2 R_{11} - R_{12}^2 R_{33}$$

$$\tilde{R}_{sw} = R_{11}R_{22} - R_{12}^2$$

$$\tilde{R}_{iw} = R_{22}R_{33} - R_{23}^2$$

$$\tilde{R}_{si} = R_{11}R_{33}$$

$$\tilde{\rho} = \rho_{11}\rho_{22}\rho_{33} - \rho_{23}^2\rho_{11} - \rho_{12}^2\rho_{33}$$

$$\tilde{\rho}_{sw} = \rho_{11}\rho_{22} - \rho_{12}^2$$

$$\tilde{\rho}_{iw} = \rho_{22}\rho_{33} - \rho_{23}^2$$

$$\tilde{\rho}_{si} = \rho_{11}\rho_{33}$$

The roots of the third degree characteristic equation, denoted as $\Lambda_1$, $\Lambda_2$ and $\Lambda_3$, can be found by computing the eigenvalues of the companion matrix (Horn and Johnson, 2012). The velocities of the three types of P-wave ($v_{p1} > v_{p2} > v_{p3}$) are given as

follows:

$$v_{p1} = \sqrt{\frac{1}{\Lambda_1}}; \quad v_{p2} = \sqrt{\frac{1}{\Lambda_2}}; \, v_{p3} = \sqrt{\frac{1}{\Lambda_3}}$$

The velocities of the two types of S-wave are determined by a second degree characteristic equation:

$$\delta^2 \rho_{22}\tilde{\mu}_{si} - \delta(\mu_{11}\tilde{\rho}_{iw} + \mu_{33}\tilde{\rho}_{sw}) + \tilde{\rho} = 0$$

The roots of this second degree characteristic equation is denoted by $\delta_1$ and $\delta_2$. The velocities of the two types of S-wave

($v_{s1} > v_{s2}$) are given as follows:

$$v_{s1} = \sqrt{\frac{1}{\delta_1}}; \quad v_{s2} = \sqrt{\frac{1}{\delta_2}};$$





## Appendix B: Inversion results for other sections

The inversion results for the sections ranging from 120 m to 600 m are summarized in Figure B.1 to Figure B.4.

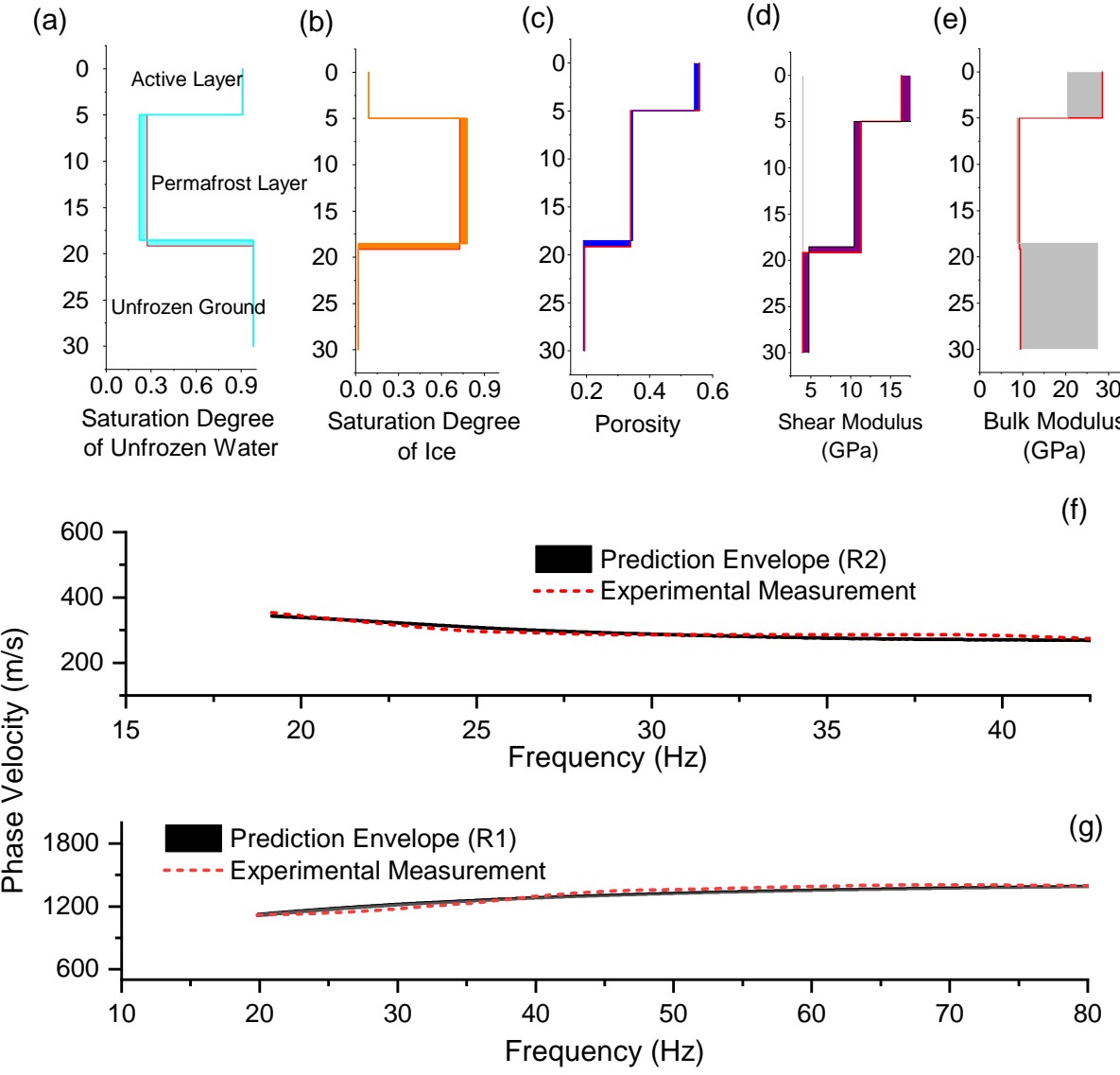

**Figure B.1.** Surface wave inversion results for Section 2: 120m to 240m. **a** Degree of saturation of unfrozen water, **b** Degree of saturation of ice, **c** Porosity distribution, **d** Shear modulus of solid skeletal frame, **e** Bulk modulus of solid skeletal frame, **f** Experimental and numerical dispersion curves for R2 wave, **g** Experimental and numerical dispersion curves for R1 wave.





**Figure B.2.** Surface wave inversion results for Section 3: 240m to 360m. **a** Degree of saturation of unfrozen water, **b** Degree of saturation of ice, **c** Porosity distribution, **d** Shear modulus of solid skeletal frame, **e** Bulk modulus of solid skeletal frame, **f** Experimental and numerical dispersion curves for R2 wave, **g** Experimental and numerical dispersion curves for R1 wave.

**Figure B.3.** Surface wave inversion results for Section 4 (from 360m to 480m). **a** Degree of saturation of unfrozen water, **b** Degree of saturation of ice, **c** Porosity distribution, **d** Shear modulus of solid skeletal frame, **e** Bulk modulus of solid skeletal frame, **f** Experimental and numerical dispersion curves for R2 wave, **g** Experimental and numerical dispersion curves for R1 wave.





**Figure B.4.** Surface wave inversion results for Section 5 (from 480m to 600m). **a** Degree of saturation of unfrozen water, **b** Degree of saturation of ice, **c** Porosity distribution, **d** Shear modulus of solid skeletal frame, **e** Bulk modulus of solid skeletal frame, **f** Experimental and numerical dispersion curves for R2 wave, **g** Experimental and numerical dispersion curves for R1 wave.





## Appendix C: Forward three-phase poromechanical model

**Kinematics assumptions**

The Green-Lagrange strain tensor ($\epsilon_{ij}$) for infinitesimal deformations expressed as displacement vector $u^1$, $u^2$ and $u^3$ for the solid skeleton, pore water and pore ice are shown in Equation C1.

$$\begin{cases} \epsilon_{ij}^1 = \frac{1}{2}(u_{i,j}^1 + u_{j,i}^1) \\ \epsilon_{ij}^2 = \frac{1}{3}\epsilon_{kk}^2 \delta_{ij} \quad (\epsilon_{kk}^2 = u_{k,k}^2) \\ \epsilon_{ij}^3 = \frac{1}{2}(u_{i,j}^3 + u_{j,i}^3) \end{cases} \tag{C1}$$

where $\delta_{ij}$ is the identity tensor.

The strain tensor of pore water $\epsilon_{ij}^2$ is diagonal since the shear deformation does not exist in pore water component.

**Constitutive model**

The constitutive models defined as the relation between the stress and strain tensors for solid skeleton, pore water and pore ice are given in Equation C2:

$$\begin{cases} \sigma_{ij}^1 = (K_1\theta_1 + C_{12}\theta_2 + C_{13}\theta_3)\delta_{ij} + 2\mu_1 d_{ij}^1 + \mu_{13} d_{ij}^3 \\ \sigma^2 = C_{12}\theta_1 + K_2\theta_2 + C_{23}\theta_3 \\ \sigma_{ij}^3 = (K_3\theta_3 + C_{23}\theta_2 + C_{13}\theta_1)\delta_{ij} + 2\mu_3 d_{ij}^3 + \mu_{13} d_{ij}^1 \end{cases} \tag{C2}$$

in which $\sigma^1$, $\sigma^2$ and $\sigma^3$ are the effective stress, pore water pressure and ice pressure, respectively. The definition of each term (e.g., $K_1$, $C_{12}$, $C_{13}$, $\mu_1$, $\mu_{13}$, $K_2$, $C_{23}$, $K_3$, $\mu_3$) in Equation C2 is given in D. The term $\theta_m$, $d_{ij}^m$ and $\epsilon_{ij}^m$ ($m$, ranging from 1 to 3, represents the different phases) are defined as follows:

$$\begin{cases} \theta_m = \epsilon_{kk}^m \\ d_{ij}^m = \epsilon_{ij}^m - \frac{1}{3}\delta_{ij}\theta_m \\ \epsilon_{ij}^m = \frac{1}{2}(u_{i,j}^m + u_{j,i}^m). \end{cases}$$

**Conservation laws**

The momentum conservation considers the acceleration of each component and the existing relative motion of the pore ice and pore water phases with respect to the solid skeleton. The momentum conservation for the three phases is given by Equation C3.

$$\begin{cases} \sigma_{ij,j}^1 = \rho_{11}\ddot{u}_i^1 + \rho_{12}\ddot{u}_i^2 + \rho_{13}\ddot{u}_i^3 - b_{12}(\dot{u}_i^2 - \dot{u}_i^1) - b_{13}(\dot{u}_i^3 - \dot{u}_i^1) \\ \sigma_{,i}^2 = \rho_{12}\ddot{u}_i^1 + \rho_{22}\ddot{u}_i^2 + \rho_{23}\ddot{u}_i^3 + b_{12}(\dot{u}_i^2 - \dot{u}_i^1) + b_{23}(\dot{u}_i^2 - \dot{u}_i^3) \\ \sigma_{ij,j}^3 = \rho_{13}\ddot{u}_i^1 + \rho_{23}\ddot{u}_i^2 + \rho_{33}\ddot{u}_i^3 - b_{23}(\dot{u}_i^2 - \dot{u}_i^1) + b_{13}(\dot{u}_i^3 - \dot{u}_i^1) \end{cases} \tag{C3}$$





in which the expressions for the density terms ($\rho_{ij}$ or $\bar{\rho}$ in matrix form) and viscous matrix ($b_{ij}$ or $\bar{b}$ in matrix form) are given

in D; $\ddot{u}$ and $\dot{u}$ represent second and first derivative of displacement vectors with respect to time; the subscript $i$ represents the

component in $r$, $\theta$ and $z$ direction in cylindrical coordinates.

Through the infinitesimal kinematic assumptions, the stress-strain constitutive model and conversation of momentum, the

field equation can be written in the matrix form, as shown in Equation C4.

$$\bar{\rho}\begin{bmatrix}\ddot{u}_i^1\\\ddot{u}_i^2\\\ddot{u}_i^3\end{bmatrix}+\bar{b}\begin{bmatrix}\dot{u}_i^1\\\dot{u}_i^2\\\dot{u}_i^3\end{bmatrix}=\bar{R}\,\nabla\nabla\cdot\begin{bmatrix}u_i^1\\u_i^2\\u_i^3\end{bmatrix}-\bar{\mu}\,\nabla\times\nabla\times\begin{bmatrix}u_i^1\\u_i^2\\u_i^3\end{bmatrix} \tag{C4}$$

in which the matrix $\bar{R}$ and $\bar{\mu}$ are given in D.

By performing divergence operation ($\nabla\cdot$) and curl operation ($\nabla\times$) on both sides of Equation C4, the field equation in the

frequency domain can be written as Equation C5.

$$\begin{cases}-\bar{\rho}\,\omega^2\,\nabla\cdot\begin{bmatrix}u_i^1\\u_i^2\\u_i^3\end{bmatrix}-\bar{b}\,i\,\omega\,\nabla\cdot\begin{bmatrix}u_i^1\\u_i^2\\u_i^3\end{bmatrix}=\bar{R}\,\nabla^2\nabla\cdot\begin{bmatrix}u_i^1\\u_i^2\\u_i^3\end{bmatrix}\\[2em]-\bar{\rho}\,\omega^2\,\nabla\times\begin{bmatrix}u_i^1\\u_i^2\\u_i^3\end{bmatrix}-\bar{b}\,i\,\omega\,\nabla\times\begin{bmatrix}u_i^1\\u_i^2\\u_i^3\end{bmatrix}=\bar{\mu}\,\nabla^2\nabla\times\begin{bmatrix}u_i^1\\u_i^2\\u_i^3\end{bmatrix}.\end{cases} \tag{C5}$$

Using the Helmholtz decomposition theorem allows us to decompose the displacement field, $\bar{u}$ (equivalent to $u_i$), into the

longitudinal potential and transverse vector components as follows:

$$\begin{cases}\bar{u}^1=\nabla\phi_1+\nabla\times\bar{\psi}_1 \quad and \quad \nabla\cdot\bar{\psi}_1=0\\\bar{u}^2=\nabla\phi_2+\nabla\times\bar{\psi}_2 \quad and \quad \nabla\cdot\bar{\psi}_2=0\\\bar{u}^3=\nabla\phi_3+\nabla\times\bar{\psi}_3 \quad and \quad \nabla\cdot\bar{\psi}_3=0.\end{cases} \tag{C6}$$

By substituting Equation C6 into the field equation of motion, Equation C5, we obtain two sets of uncoupled partial differ-

ential equations relative to the compressional wave P related to the Helmholtz scalar potentials , and to the shear wave S related

to the Helmholtz vector potential, respectively (Equation C7). In the axi-symmetric condition, only the second components

exits in vector $\bar{\psi}$, which is denoted as $\psi$ in the future. It should be mentioned that the field equations in Laplace domain can be





easily obtained by replacing $\omega$ with $i.s$ ($i^2 = -1$ and $s$ the Laplace variable).

$$
\begin{cases}
-\bar{\rho}\,\omega^2 \begin{bmatrix} \phi_1 \\ \phi_2 \\ \phi_3 \end{bmatrix} - \bar{b}\,i\,\omega \begin{bmatrix} \phi_1 \\ \phi_2 \\ \phi_3 \end{bmatrix} = \bar{R}\,\nabla^2 \begin{bmatrix} \phi_1 \\ \phi_2 \\ \phi_3 \end{bmatrix} \\[3em]
-\bar{\rho}\,\omega^2 \begin{bmatrix} \psi_1 \\ \psi_2 \\ \psi_3 \end{bmatrix} - \bar{b}\,i\,\omega \begin{bmatrix} \psi_1 \\ \psi_2 \\ \psi_3 \end{bmatrix} = \bar{\mu}\,\nabla^2 \begin{bmatrix} \psi_1 \\ \psi_2 \\ \psi_3 \end{bmatrix}.
\end{cases}
\tag{C7}
$$

**Solution for the longitudinal waves (P waves) by eigen decomposition**

Equation (C7) shows that $\phi_1$, $\phi_2$ and $\phi_3$ are coupled in the field equations. The diagonalization of such a matrix is required to decouple the system. Equation (C7) is then rearranged into Equation (C8):

$$
\nabla^2 \begin{bmatrix} \phi_1 \\ \phi_2 \\ \phi_3 \end{bmatrix} = \underbrace{-\bar{R}^{-1}(\bar{\rho}\omega^2 + \bar{b}\,i\,\omega)}_{\bar{K}} \begin{bmatrix} \phi_1 \\ \phi_2 \\ \phi_3 \end{bmatrix}
\tag{C8}
$$

where the $\bar{K}$ matrix can be rewritten using the Eigen decomposition:

$$
\bar{K} = \bar{P}\,\bar{D}\,\bar{P}^{-1}
\tag{C9}
$$

where $\bar{P}$ is the eigenvector and $\bar{D}$ is the eigenvalue matrix of $\bar{K}$.

By setting $\bar{\phi} = \bar{P}\bar{y}$, where $\bar{y} = [\phi_{p1}, \phi_{p2}, \phi_{p3}]$, we can obtain $\nabla^2\bar{y} = \bar{D}\bar{y}$. The equation of longitudinal wave has been decoupled. In cylindrical coordinates, the solution for $\bar{y} = [\phi_{p1}, \phi_{p2}, \phi_{p3}]$ is summarized as follows:

$$
\begin{cases}
\phi_{p1}(r,z) = A e^{-\sqrt{k^2 + D_{11}}\,z} J_0(k\,r) \\
\phi_{p2}(r,z) = B e^{-\sqrt{k^2 + D_{22}}\,z} J_0(k\,r) \\
\phi_{p3}(r,z) = C e^{-\sqrt{k^2 + D_{33}}\,z} J_0(k\,r)
\end{cases}
\tag{C10}
$$

where $k$ is the wave number; coefficient $A$, $B$ and $C$ will be determined by boundary conditions; $D_{11}$, $D_{22}$, and $D_{33}$ are the diagonal components of $\bar{D}$; $J_0$ is the Bessel function of the first kind. For simplicity, The terms $\sqrt{k^2 + D_{11}}$, $\sqrt{k^2 + D_{22}}$ and $\sqrt{k^2 + D_{33}}$ are denoted as $k_{p1}$, $k_{p2}$ and $k_{p3}$, respectively.

Now, the P wave potentials can be written as:

$$
\begin{Bmatrix} \phi_s \\ \phi_w \\ \phi_i \end{Bmatrix} = \begin{Bmatrix} p_{11} & p_{12} & p_{13} \\ p_{21} & p_{22} & p_{23} \\ p_{31} & p_{32} & p_{33} \end{Bmatrix} \begin{Bmatrix} \phi_{p1} \\ \phi_{p2} \\ \phi_{p3} \end{Bmatrix}
\tag{C11}
$$

where $p_{ij}$ are the components for the eigenvector of $\bar{P}$.



## Solution for shear waves (S waves)

The solutions for the S wave potentials can be solved in a similar manner. The Equation C12 is firstly rearranged into Equation C13:

$$-\bar{\rho}\,\omega^2 \begin{bmatrix} \psi_s \\ \psi_w \\ \psi_i \end{bmatrix} - \bar{b}\,i\,\omega \begin{bmatrix} \psi_s \\ \psi_w \\ \psi_i \end{bmatrix} = \bar{\mu}\,\nabla^2 \begin{bmatrix} \psi_s \\ \psi_w \\ \psi_i \end{bmatrix} \tag{C12}$$

$$\underbrace{-\bar{\rho}\omega^2 - \bar{b}\,i\,\omega}_{\bar{A}} \begin{bmatrix} \psi_s \\ \psi_w \\ \psi_i \end{bmatrix} = \bar{\mu}\,\nabla^2 \begin{bmatrix} \psi_s \\ \psi_w \\ \psi_i \end{bmatrix} \tag{C13}$$

where the matrix $\bar{A}$ is given in D.

Since $\psi_w$ can be expressed as a function of $\psi_s$ and $\psi_i$ (shown in Equation C14), the Equation C13 is further simplified and rearranged into Equation C15.

$$\begin{cases} A_{21}\psi_s + A_{22}\psi_w + A_{23}\psi_i = 0 \\ \psi_w = -\frac{A_{21}\psi_s + A_{23}\psi_i}{A_{22}} \end{cases} \tag{C14}$$

$$\nabla^2 \begin{bmatrix} \psi_s \\ \psi_i \end{bmatrix} = \underbrace{\begin{bmatrix} \mu_{11} & \mu_{13} \\ \mu_{13} & \mu_{33} \end{bmatrix}^{-1} \bar{C}}_{\bar{N}} \begin{bmatrix} \psi_s \\ \psi_i \end{bmatrix}. \tag{C15}$$

where

$$\bar{C} = \begin{pmatrix} A_{11} - \frac{A_{12}A_{21}}{A_{22}} & A_{13} - \frac{A_{12}A_{23}}{A_{22}} \\ A_{31} - \frac{A_{32}A_{21}}{A_{22}} & A_{33} - \frac{A_{32}A_{23}}{A_{22}} \end{pmatrix}$$

The $\bar{N}$ matrix can be rewritten using the eigen decomposition ($\bar{N} = \bar{Q}\,\bar{G}\,\bar{Q}^{-1}$), where $\bar{Q}$ is the eigenvector and $\bar{G}$ is the

eigenvalue matrix of $\bar{N}$. By setting $\bar{\psi} = \bar{Q}\,\bar{y}'$ where $\bar{y}' = [\psi_{s1}, \psi_{i1}]$, we can obtain:

$$\psi_{s1} = E e^{-\sqrt{k^2 + G_{11}}\,z} J_1(k\,r) \tag{C16}$$

$$\psi_{i1} = F e^{-\sqrt{k^2 + G_{22}}\,z} J_1(k\,r) \tag{C17}$$

where $J_1$ is the Bessel function of the first kind with order 1. $G_{11}$ and $G_{22}$ are the diagonal components of matrix $\bar{G}$. For simplicity, the term $\sqrt{k^2 + G_{11}}$ and $\sqrt{k^2 + G_{22}}$ is denoted as $k_{s1}$ and $k_{s2}$.

Finally, the solution of the S wave potentials can be written as:

$$\begin{Bmatrix} \psi_s \\ \psi_i \end{Bmatrix} = \begin{Bmatrix} Q_{11} & Q_{12} \\ Q_{21} & Q_{22} \end{Bmatrix} \begin{Bmatrix} \psi_{s1} \\ \psi_{i1} \end{Bmatrix} \tag{C18}$$

where $Q_{ij}$ are the components for eigenvector of $\bar{Q}$.



**Layer element with finite thickness**

By including both incident wave and reflected wave, the potentials for a layer with finite thickness can be written in Equation
C19:

$$
\begin{bmatrix} u_{r1}^1 \\ u_{z1}^1 \\ u_{z1}^2 \\ u_{r1}^3 \\ u_{z1}^3 \\ u_{r2}^1 \\ u_{z2}^1 \\ u_{z2}^2 \\ u_{r2}^3 \\ u_{z2}^3 \end{bmatrix} = \begin{bmatrix} & & \\ & S_1 & \\ & & \end{bmatrix} \begin{bmatrix} A_1 \\ B_1 \\ C_1 \\ E_1 \\ F_1 \\ A_2 \\ B_2 \\ C_2 \\ E_2 \\ F_2 \end{bmatrix} \tag{C19}
$$

where the components of $S_1$ is given in E; the subscript 1 and 2 represent node for the upper and lower layer, respectively. The
coefficient $A$ to $F$ is determined by the boundary condition.

The matrix of effective stress, pore water pressure and pore ice pressure in the frequency domain is shown in Equation C20
in which the components for matrix $S_2$ can be found in the E.

$$
\begin{bmatrix} \sigma_{r1}^1 \\ \sigma_{z1}^1 \\ p_1 \\ \sigma_{r1}^3 \\ \sigma_{z1}^3 \\ \sigma_{r2}^1 \\ \sigma_{z2}^1 \\ p_2 \\ \sigma_{r2}^3 \\ \sigma_{z2}^3 \end{bmatrix} = \begin{bmatrix} & & \\ & S_2 & \\ & & \end{bmatrix} \begin{bmatrix} A_1 \\ B_1 \\ C_1 \\ E_1 \\ F_1 \\ A_2 \\ B_2 \\ C_2 \\ E_2 \\ F_2 \end{bmatrix}. \tag{C20}
$$

According to the Cauchy stress principle, the traction force ($T$) is taken as the dot product between the stress tensor and the
unit vector along the outward normal direction. Due to the convection that the upward direction is negative, the upper boundary
becomes negative. Similarly, to make the sign consistent, the $N$ matrix is applied to matrix $S_2 \cdot S_1^{-1}$. In the future, the matrix





$N \cdot S_2 \cdot S_1^{-1}$ will be denoted as the $G_i$ matrix, in which $i$ denotes the layer number.

$$
\begin{bmatrix} T_{r1}^1 \\ T_{z1}^1 \\ T_1 \\ T_{r1}^3 \\ T_{z1}^3 \\ T_{r2}^1 \\ T_{z2}^1 \\ T_2 \\ T_{r2}^3 \\ T_{z2}^3 \end{bmatrix}_i = \begin{bmatrix} -\sigma_{r1}^1 \\ -\sigma_{z1}^1 \\ -p_1 \\ -\sigma_{r1}^3 \\ -\sigma_{z1}^3 \\ \sigma_{r2}^1 \\ \sigma_{z2}^1 \\ p_2 \\ \sigma_{r2}^3 \\ \sigma_{z2}^3 \end{bmatrix}_i = \underbrace{N \cdot S_2 \cdot S_1^{-1}}_{G_i} \cdot \begin{bmatrix} u_{r1}^1 \\ u_{z1}^1 \\ u_{z1}^2 \\ u_{r1}^3 \\ u_{z1}^3 \\ u_{r2}^1 \\ u_{z2}^1 \\ u_{z2}^2 \\ u_{r2}^3 \\ u_{z2}^3 \end{bmatrix}_i
$$
(C21)

where

$$
N = \begin{bmatrix} -1 & 0 & 0 & 0 & 0 & 0 & 0 & 0 & 0 & 0 \\ 0 & -1 & 0 & 0 & 0 & 0 & 0 & 0 & 0 & 0 \\ 0 & 0 & -1 & 0 & 0 & 0 & 0 & 0 & 0 & 0 \\ 0 & 0 & 0 & -1 & 0 & 0 & 0 & 0 & 0 & 0 \\ 0 & 0 & 0 & 0 & -1 & 0 & 0 & 0 & 0 & 0 \\ 0 & 0 & 0 & 0 & 0 & 1 & 0 & 0 & 0 & 0 \\ 0 & 0 & 0 & 0 & 0 & 0 & 1 & 0 & 0 & 0 \\ 0 & 0 & 0 & 0 & 0 & 0 & 0 & 1 & 0 & 0 \\ 0 & 0 & 0 & 0 & 0 & 0 & 0 & 0 & 1 & 0 \\ 0 & 0 & 0 & 0 & 0 & 0 & 0 & 0 & 0 & 1 \end{bmatrix}.
$$
(C22)

**Layer element with infinite thickness**

By assuming that no wave reflects back to a semi-infinite element, one-node element with infinite thickness is applied. The matrix for the displacement components in one-node layer are written as Equation C23. The matrix $S_1$ is reduced into a 5 by 5 matrix ($S_{1ij}$ where $i$ and $j$ range from 1 to 5). The value of each components are shown in E.

$$
\begin{bmatrix} u_{r1}^1 \\ u_{z1}^1 \\ u_{z1}^2 \\ u_{r1}^3 \\ u_{z1}^3 \end{bmatrix} = \begin{bmatrix} & & \\ & S_1 & \\ & & \end{bmatrix} \begin{bmatrix} A_1 \\ B_1 \\ C_1 \\ E_1 \\ F_1 \end{bmatrix}.
$$
(C23)

Similarly, the matrix of effective stress components and porewater pressure in the frequency domain is shown in Equation C24. The matrix $S_2$ is reduced into a 5 by 5 matrix ($S_{2ij}$ where $i$ and $j$ range from 1 to 5). The matrix $G_h$ in Figure 2 is





calculated as $G_h = S_2\, S_1^{-1}$. The value of each components are shown in E.

$$
\begin{bmatrix} \sigma_{r1}^1 \\ \sigma_{z1}^1 \\ p_1 \\ \sigma_{r1}^3 \\ \sigma_{z1}^3 \end{bmatrix} = \begin{bmatrix} & & S_2 & & \end{bmatrix} \begin{bmatrix} A_1 \\ B_1 \\ C_1 \\ E_1 \\ F_1 \end{bmatrix}. \tag{C24}
$$





## Appendix D: Parameters definition

The matrix $\bar{\rho}$, $\bar{b}$, $\bar{R}$, $\bar{\mu}$ and $\bar{A}$ are defined as follows:

$$\bar{\rho} = \begin{bmatrix} \rho_{11} & \rho_{12} & \rho_{13} \\ \rho_{12} & \rho_{22} & \rho_{23} \\ \rho_{13} & \rho_{23} & \rho_{33} \end{bmatrix} \quad \bar{b} = \begin{bmatrix} b_{12}+b_{13} & -b_{12} & -b_{13} \\ -b_{12} & b_{12}+b_{23} & -b_{23} \\ -b_{13} & -b_{23} & b_{13}+b_{23} \end{bmatrix}$$

$$\bar{R} = \begin{bmatrix} R_{11} & R_{12} & R_{13} \\ R_{12} & R_{22} & R_{23} \\ R_{13} & R_{23} & R_{33} \end{bmatrix} \quad \bar{\mu} = \begin{bmatrix} \mu_{11} & 0 & \mu_{13} \\ 0 & 0 & 0 \\ \mu_{13} & 0 & \mu_{33} \end{bmatrix}$$

$$\bar{A} = - \begin{pmatrix} \omega((b_{12}+b_{13})i + \rho_{11}\omega) & \omega(\rho_{12}\omega - b_{12}i) & \omega(\rho_{13}\omega - b_{13}i) \\ \omega(\rho_{12}\omega - b_{12}i) & \omega((b_{12}+b_{23})i + \rho_{22}\omega) & \omega(\rho_{23}\omega - b_{23}i) \\ \omega(\rho_{13}\omega - b_{13}i) & \omega(\rho_{23}\omega - b_{23}i) & \omega((b_{13}+b_{23})i + \rho_{33}\omega) \end{pmatrix}.$$

$$a_{12} = r_{12}\frac{\phi_s(\phi_w\rho_w + \phi_i\rho_i)}{\phi_w\rho_w(\phi_w + \phi_i)} + 1$$

$$a_{23} = r_{23}\frac{\phi_s(\phi_w\rho_w + \phi_s\rho_s)}{\phi_w\rho_w(\phi_w + \phi_s)} + 1$$

$$a_{13} = r_{13}\frac{\phi_i(\phi_s\rho_s + \phi_i\rho_i)}{\phi_s\rho_s(\phi_s + \phi_i)} + 1$$

$$a_{31} = r_{31}\frac{\phi_s(\phi_s\rho_s + \phi_i\rho_i)}{\phi_i\rho_i(\phi_s + \phi_i)} + 1$$

$$\rho_{11} = a_{13}\phi_s\rho_s + (a_{12}-1)\phi_w\rho_w + (a_{31}-1)\phi_i\rho_i$$

$$\rho_{22} = (a_{12} + a_{23} - 1)\phi_w\rho_w$$

$$\rho_{33} = (a_{13}-1)\phi_s\rho_s + (a_{23}-1)\phi_w\rho_w + a_{31}\phi_i\rho_i$$

$$\rho_{12} = -(a_{12}-1)\phi_w\rho_w$$

$$\rho_{13} = -(a_{13}-1)\phi_s\rho_s - (a_{31}-1)\phi_i\rho_i \quad \rho_{23} = -(a_{23}-1)\phi_w\rho_w$$

$b_{12} = \eta_w\phi_w^2/\kappa_s$ :friction coefficient between the solid skeletal frame and pore water

$b_{23} = \eta_w\phi_w^2/\kappa_i$ :friction coefficient between pore water and ice matrix

$b_{13} = b_{13}^0(\phi_i\phi_s)^2$ :friction coefficient between the solid skeletal frame and ice matrix

$$\kappa_s = \kappa_{s0}s_r^3$$

$$\kappa_i = \kappa_{i0}\phi^3/[(1-s_r^2)(1-\phi)^3]$$

$$R_{11} = [(1-c_1)\phi_s]^2 K_{av} + K_{sm} + 4\mu_{11}/3$$

$$R_{22} = \phi_w^2 K_{av}$$

$$R_{33} = [(1-c_3)\phi_i]^2 K_{av} + K_{im} + 4\mu_{33}/3$$

$$R_{12} = (1-c_1)\phi_s\phi_w K_{av}$$

$$R_{13} = (1-c_1)(1-c_3)\phi_s\phi_i K_{av} + 2\mu_{13}/3$$





$$R_{23} = (1 - c_3)\phi_w\phi_i K_{av}$$

$$\mu_{11} = [(1 - g_1)\phi_s]^2\mu_{av} + \mu_{sm}$$

$$\mu_{33} = [(1 - g_3)\phi_i]^2\mu_{av} + \mu_{im}$$

$$\mu_{13} = (1 - g_1)(1 - g_3)\mu_{av}$$

$c_1 = K_{sm}/(\phi_s K_s)$ :consolidation coefficient for the solid skeletal frame

$c_3 = K_{im}/(\phi_i K_i)$ :consolidation coefficient for the ice

$$g_1 = \mu_{sm}/(\phi_s\mu_s)$$

$$g_3 = \mu_{im}/(\phi_i\mu_i)$$

$K_{im} = \phi_i K_i/[1 + \alpha(1 - \phi_i)]$ :bulk modulus of the matrix formed by the ice

$\mu_{im} = \phi_i\mu_i/[1 + \alpha\gamma(1 - \phi_i)]$ :shear modulus of the matrix formed by the ice

$K_{sm} = (1 - \phi_w - \bar{\xi}\phi_i)K_s/[1 + \alpha(\phi_w + \bar{\xi}\phi_i)]$ :bulk modulus of the matrix formed by the solid skeletal frame

$\mu_{sm} = (1 - \phi_w - \bar{\xi}\phi_i)\mu_s/[1 + \alpha\gamma(\phi_w + \bar{\xi}\phi_i)]$ :shear modulus of the solid skeletal frame

$$Sc_2 = C_{13} - \tfrac{1}{3}\mu_{13}$$

$$Sc_3 = K_3 - \tfrac{2}{3}\mu_3$$

$$Sc_4 = C_{13} - \tfrac{1}{3}\mu_{13}$$

$$K_1 = [(1 - c_1)\phi_s]^2 K_{av} + K_{sm}$$

$$K_3 = [(1 - c_3)\phi_i]^2 K_{av} + K_{im}$$





## Appendix E: Spectral element matrix components

The components of the $S_1$ matrix in the Equation C19 are shown as follows:

$S_1(1,1) = -kp_{11}$  $\qquad$  $S_1(1,2) = -kp_{12}$

$S_1(1,3) = -kp_{13}$  $\qquad$  $S_1(1,4) = k_{s1}q_{11}$

$S_1(1,5) = k_{s2}q_{12}$  $\qquad$  $S_1(1,6) = kp_{11}\left(-e^{-hk_{p1}}\right)$

$S_1(1,7) = kp_{12}\left(-e^{-hk_{p2}}\right)$  $\qquad$  $S_1(1,8) = kp_{13}\left(-e^{-hk_{p3}}\right)$

$S_1(1,9) = k_{s1}q_{11}\left(-e^{-hk_{s1}}\right)$  $\qquad$  $S_1(1,10) = k_{s2}q_{12}\left(-e^{-hk_{s2}}\right)$

$S_1(2,1) = -k_{p1}p_{11}$  $\qquad$  $S_1(2,2) = -k_{p2}p_{12}$

$S_1(2,3) = -k_{p3}p_{13}$  $\qquad$  $S_1(2,4) = kq_{11}$

$S_1(2,5) = kq_{12}$  $\qquad$  $S(2,6) = e^{-hk_{p1}}k_{p1}p_{11}$

$S_1(2,7) = e^{-hk_{p2}}k_{p2}p_{12}$  $\qquad$  $S_1(2,8) = e^{-hk_{p3}}k_{p3}p_{13}$

$S_1(2,9) = e^{-hk_{s1}}kq_{11}$  $\qquad$  $S_1(2,10) = e^{-hk_{s2}}kq_{12}$

$S_1(3,1) = -k_{p1}p_{21}$  $\qquad$  $S(3,2) = -k_{p2}p_{22}$

$S_1(3,3) = -k_{p3}p_{23}$  $\qquad$  $S_1(3,4) = k(G_1q_{11} + G_2q_{21})$

$S_1(3,5) = k(G_1q_{12} + G_2q_{22})$  $\qquad$  $S_1(3,6) = e^{-hk_{p1}}k_{p1}p_{21}$

$S_1(3,7) = e^{-hk_{p2}}k_{p2}p_{22}$  $\qquad$  $S_1(3,8) = e^{-hk_{p3}}k_{p3}p_{23}$

$S_1(3,9) = e^{-hk_{s1}}k(G_1q_{11} + G_2q_{21})$  $\qquad$  $S_1(3,10) = e^{-hk_{s2}}k(G_1q_{12} + G_2q_{22})$

$S_1(4,1) = -k_{p1}p_{21}$  $\qquad$  $S(4,2) = -k_{p2}p_{22}$

$S_1(4,3) = -k_{p3}p_{23}$  $\qquad$  $S_1(4,4) = k(G_1q_{11} + G_2q_{21})$

$S_1(4,5) = k(G_1q_{12} + G_2q_{22})$  $\qquad$  $S_1(4,6) = e^{-hk_{p1}}k_{p1}p_{21}$

$S_1(4,7) = e^{-hk_{p2}}k_{p2}p_{22}$  $\qquad$  $S_1(4,8) = e^{-hk_{p3}}k_{p3}p_{23}$

$S_1(4,9) = e^{-hk_{s1}}k(G_1q_{11} + G_2q_{21})$  $\qquad$  $S_1(4,10) = e^{-hk_{s2}}k(G_1q_{12} + G_2q_{22})$

$S_1(5,1) = -k_{p1}p_{21}$  $\qquad$  $S(5,2) = -k_{p2}p_{22}$

$S_1(5,3) = -k_{p3}p_{23}$  $\qquad$  $S_1(5,4) = k(G_1q_{11} + G_2q_{21})$

$S_1(5,5) = k(G_1q_{12} + G_2q_{22})$  $\qquad$  $S_1(5,6) = e^{-hk_{p1}}k_{p1}p_{21}$

$S_1(5,7) = e^{-hk_{p2}}k_{p2}p_{22}$  $\qquad$  $S_1(5,8) = e^{-hk_{p3}}k_{p3}p_{23}$

$S_1(5,9) = e^{-hk_{s1}}k(G_1q_{11} + G_2q_{21})$  $\qquad$  $S_1(5,10) = e^{-hk_{s2}}k(G_1q_{12} + G_2q_{22})$





$S_1(6,1) = -k_{p1}p_{21}$      $S(6,2) = -k_{p2}p_{22}$

$S_1(6,3) = -k_{p3}p_{23}$      $S_1(6,4) = k(G_1q_{11} + G_2q_{21})$

$S_1(6,5) = k(G_1q_{12} + G_2q_{22})$      $S_1(6,6) = e^{-hk_{p1}}k_{p1}p_{21}$

$S_1(6,7) = e^{-hk_{p2}}k_{p2}p_{22}$      $S_1(6,8) = e^{-hk_{p3}}k_{p3}p_{23}$

$S_1(6,9) = e^{-hk_{s1}}k(G_1q_{11} + G_2q_{21})$      $S_1(6,10) = e^{-hk_{s2}}k(G_1q_{12} + G_2q_{22})$

<br>

$S_1(7,1) = -k_{p1}p_{21}$      $S(7,2) = -k_{p2}p_{22}$

$S_1(7,3) = -k_{p3}p_{23}$      $S_1(7,4) = k(G_1q_{11} + G_2q_{21})$

$S_1(7,5) = k(G_1q_{12} + G_2q_{22})$      $S_1(7,6) = e^{-hk_{p1}}k_{p1}p_{21}$

$S_1(7,7) = e^{-hk_{p2}}k_{p2}p_{22}$      $S_1(7,8) = e^{-hk_{p3}}k_{p3}p_{23}$

$S_1(7,9) = e^{-hk_{s1}}k(G_1q_{11} + G_2q_{21})$      $S_1(7,10) = e^{-hk_{s2}}k(G_1q_{12} + G_2q_{22})$

<br>

$S_1(8,1) = -k_{p1}p_{21}$      $S(8,2) = -k_{p2}p_{22}$

$S_1(8,3) = -k_{p3}p_{23}$      $S_1(8,4) = k(G_1q_{11} + G_2q_{21})$

$S_1(8,5) = k(G_1q_{12} + G_2q_{22})$      $S_1(8,6) = e^{-hk_{p1}}k_{p1}p_{21}$

$S_1(8,7) = e^{-hk_{p2}}k_{p2}p_{22}$      $S_1(8,8) = e^{-hk_{p3}}k_{p3}p_{23}$

$S_1(8,9) = e^{-hk_{s1}}k(G_1q_{11} + G_2q_{21})$      $S_1(8,10) = e^{-hk_{s2}}k(G_1q_{12} + G_2q_{22})$

<br>

$S_1(9,1) = -k_{p1}p_{21}$      $S(9,2) = -k_{p2}p_{22}$

$S_1(9,3) = -k_{p3}p_{23}$      $S_1(9,4) = k(G_1q_{11} + G_2q_{21})$

$S_1(9,5) = k(G_1q_{12} + G_2q_{22})$      $S_1(9,6) = e^{-hk_{p1}}k_{p1}p_{21}$

$S_1(9,7) = e^{-hk_{p2}}k_{p2}p_{22}$      $S_1(9,8) = e^{-hk_{p3}}k_{p3}p_{23}$

$S_1(9,9) = e^{-hk_{s1}}k(G_1q_{11} + G_2q_{21})$      $S_1(9,10) = e^{-hk_{s2}}k(G_1q_{12} + G_2q_{22})$

<br>

$S_1(10,1) = -k_{p1}p_{21}$      $S(10,2) = -k_{p2}p_{22}$

$S_1(10,3) = -k_{p3}p_{23}$      $S_1(10,4) = k(G_1q_{11} + G_2q_{21})$

$S_1(10,5) = k(G_1q_{12} + G_2q_{22})$      $S_1(10,6) = e^{-hk_{p1}}k_{p1}p_{21}$

$S_1(10,7) = e^{-hk_{p2}}k_{p2}p_{22}$      $S_1(10,8) = e^{-hk_{p3}}k_{p3}p_{23}$

$S_1(10,9) = e^{-hk_{s1}}k(G_1q_{11} + G_2q_{21})$      $S_1(10,10) = e^{-hk_{s2}}k(G_1q_{12} + G_2q_{22})$

The components of the $S_2$ stress matrix in the Equation C20 are shown as follows:





$$S_2(1,1) = kk_{p1}(2p_{11}\mu_1 + p_{31}\mu_{13})$$
$$S_2(1,2) = kk_{p2}(2p_{12}\mu_1 + p_{32}\mu_{13})$$
$$S_2(1,3) = kk_{p3}(2p_{13}\mu_1 + p_{33}\mu_{13})$$
$$S_2(1,4) = -\frac{1}{2}\left(k^2 + k_{s1}^2\right)(2q_{11}\mu_1 + q_{21}\mu_{13})$$
$$S_2(1,5) = -\frac{1}{2}\left(k^2 + k_{s2}^2\right)(2q_{12}\mu_1 + q_{22}\mu_{13})$$
$$S_2(1,6) = -e^{-hk_{p1}}kk_{p1}(2p_{11}\mu_1 + p_{31}\mu_{13})$$
$$S_2(1,7) = e^{-hk_{p2}}kk_{p2}(2p_{12}\mu_1 + p_{32}\mu_{13})$$
$$S_2(1,8) = -e^{-hk_{p3}}kk_{p3}(2p_{13}\mu_1 + p_{33}\mu_{13})$$
$$S_2(1,9) = -\frac{1}{2}e^{-hk_{s1}}\left(k^2 + k_{s1}^2\right)(2q_{11}\mu_1 + q_{21}\mu_{13})$$
$$S_2(1,10) = -\frac{1}{2}e^{-hk_{s2}}\left(k^2 + k_{s2}^2\right)(2q_{12}\mu_1 + q_{22}\mu_{13})$$

$$S_2(2,1) = -(p_{11}S_{c1} + p_{31}S_{c2})k^2 + C_{12}\left(k_{p1}^2 - k^2\right)p_{21} + k_{p1}^2(p_{11}(S_{c1} + 2\mu_1) + p_{31}(S_{c2} + \mu_{13}))$$
$$S_2(2,2) = -(p_{12}S_{c1} + p_{32}S_{c2})k^2 + C_{12}\left(k_{p2}^2 - k^2\right)p_{22} + k_{p2}^2(p_{12}(S_{c1} + 2\mu_1) + p_{32}(S_{c2} + \mu_{13}))$$
$$S_2(2,3) = -(p_{13}S_{c1} + p_{33}S_{c2})k^2 + C_{12}\left(k_{p3}^2 - k^2\right)p_{23} + k_{p3}^2(p_{13}(S_{c1} + 2\mu_1) + p_{33}(S_{c2} + \mu_{13}))$$
$$S_2(2,4) = kk_{s1}(2q_{11}\mu_1 + q_{21}\mu_{13})$$
$$S_2(2,5) = kk_{s2}(2q_{12}\mu_1 + q_{22}\mu_{13})$$
$$S_2(2,6) = e^{-hk_{p1}}\left(-(p_{11}S_{c1} + p_{31}S_{c2})k^2 + C_{12}\left(k_{p1}^2 - k^2\right)p_{21} + k_{p1}^2(p_{11}(S_{c1} + 2\mu_1) + p_{31}(S_{c2} + \mu_{13}))\right)$$
$$S_2(2,7) = e^{-hk_{p2}}\left(-(p_{12}S_{c1} + p_{32}S_{c2})k^2 + C_{12}\left(k_{p2}^2 - k^2\right)p_{22} + k_{p2}^2(p_{12}(S_{c1} + 2\mu_1) + p_{32}(S_{c2} + \mu_{13}))\right)$$
$$S_2(2,8) = e^{-hk_{p3}}\left(-(p_{13}S_{c1} + p_{33}S_{c2})k^2 + C_{12}\left(k_{p3}^2 - k^2\right)p_{23} + k_{p3}^2(p_{13}(S_{c1} + 2\mu_1) + p_{33}(S_{c2} + \mu_{13}))\right)$$
$$S_2(2,9) = e^{-hk_{s1}}kk_{s1}(2q_{11}\mu_1 + q_{21}\mu_{13})$$
$$S_2(2,10) = e^{-hk_{s2}}kk_{s2}(2q_{12}\mu_1 + q_{22}\mu_{13})$$

$$S_2(3,1) = (k_{p1} - k)(k + k_{p1})(C_{12}p_{11} + k_2p_{21} + C_{23}p_{31})$$
$$S_2(3,2) = -(k - k_{p2})(k + k_{p2})(C_{12}p_{12} + k_2p_{22} + C_{23}p_{32})$$
$$S_2(3,3) = -(k - k_{p3})(k + k_{p3})(C_{12}p_{13} + k_2p_{23} + C_{23}P_{33})$$
$$S_2(3,4) = 0$$
$$S_2(3,5) = 0$$
$$S_2(3,6) = e^{-hk_{p1}}(k_{p1} - k)(k + k_{p1})(C_{12}p_{11} + k_2p_{21} + C_{23}p_{31})$$
$$S_2(3,7) = e^{-hk_{p2}}(k_{p2} - k)(k + k_{p2})(C_{12}p_{12} + k_2p_{22} + C_{23}p_{32})$$
$$S_2(3,8) = e^{-hk_{p3}}(k_{p3} - k)(k + k_{p3})(C_{12}p_{13} + k_2p_{23} + C_{23}P_{33})$$
$$S_2(3,9) = 0$$
$$S_2(3,10) = 0$$



$$S_2(4,1) = kk_{p1}(p_{11}\mu_{13} + 2p_{31}\mu_3)$$
$$S_2(4,2) = kk_{p2}(p_{12}\mu_{13} + 2p_{32}\mu_3)$$
$$S_2(4,3) = kk_{p3}(p_{13}\mu_{13} + 2P_{33}\mu_3)$$
$$S_2(4,4) = -\tfrac{1}{2}\left(k^2 + k_{s1}^2\right)(q_{11}\mu_{13} + 2q_{21}\mu_3)$$
$$S_2(4,5) = -\tfrac{1}{2}\left(k^2 + k_{s2}^2\right)(q_{12}\mu_{13} + 2q_{22}\mu_3)$$
$$S_2(4,6) = -e^{-hk_{p1}}kk_{p1}(p_{11}\mu_{13} + 2p_{31}\mu_3)$$
$$S_2(4,7) = -e^{-hk_{p2}}kk_{p2}(p_{12}\mu_{13} + 2p_{32}\mu_3)$$
$$S_2(4,8) = -e^{-hk_{p3}}kk_{p3}(p_{13}\mu_{13} + 2P_{33}\mu_3)$$
$$S_2(4,9) = -\tfrac{1}{2}e^{-hk_{s1}}\left(k^2 + k_{s1}^2\right)(q_{11}\mu_{13} + 2q_{21}\mu_3)$$
$$S_2(4,10) = -\tfrac{1}{2}e^{-hk_{s2}}\left(k^2 + k_{s2}^2\right)(q_{12}\mu_{13} + 2q_{22}\mu_3)$$

$$S_2(5,1) = -(p_{31}S_{c3} + p_{11}S_{c4})k^2 + C_{23}\left(k_{p1}^2 - k^2\right)p_{21} + k_{p1}^2(p_{11}(S_{c4} + \mu_{13}) + p_{31}(S_{c3} + 2\mu_3))$$
$$S_2(5,2) = -(p_{32}S_{c3} + p_{12}S_{c4})k^2 + C_{23}\left(k_{p2}^2 - k^2\right)p_{22} + k_{p2}^2(p_{12}(S_{c4} + \mu_{13}) + p_{32}(S_{c3} + 2\mu_3))$$
$$S_2(5,3) = -(P_{33}S_{c3} + p_{13}S_{c4})k^2 + C_{23}\left(k_{p3}^2 - k^2\right)p_{23} + k_{p3}^2(p_{13}(S_{c4} + \mu_{13}) + p_{33}(S_{c3} + 2\mu_3))$$
$$S_2(5,4) = -kk_{s1}(q_{11}\mu_{13} + 2q_{21}\mu_3)$$
$$S_2(5,5) = -kk_{s2}(q_{12}\mu_{13} + 2q_{22}\mu_3)$$
$$S_2(5,6) = e^{-hk_{p1}}\left(-(p_{31}S_{c3} + p_{11}S_{c4})k^2 + C_{23}\left(k_{p1}^2 - k^2\right)p_{21} + k_{p1}^2(p_{11}(S_{c4} + \mu_{13}) + p_{31}(S_{c3} + 2\mu_3))\right)$$
$$S_2(5,7) = e^{-hk_{p2}}\left(-(p_{32}S_{c3} + p_{12}S_{c4})k^2 + C_{23}\left(k_{p2}^2 - k^2\right)p_{22} + k_{p2}^2(p_{12}(S_{c4} + \mu_{13}) + p_{32}(S_{c3} + 2\mu_3))\right)$$
$$S_2(5,8) = e^{-hk_{p3}}\left(-(P_{33}S_{c3} + p_{13}S_{c4})k^2 + C_{23}\left(k_{p3}^2 - k^2\right)p_{23} + k_{p3}^2(p_{13}(S_{c4} + \mu_{13}) + p_{33}(S_{c3} + 2\mu_3))\right)$$
$$S_2(5,9) = e^{-hk_{s1}}kk_{s1}(q_{11}\mu_{13} + 2q_{21}\mu_3)$$
$$S_2(5,10) = e^{-hk_{s2}}kk_{s2}(q_{12}\mu_{13} + 2q_{22}\mu_3)$$

$$S_2(6,1) = kk_{p1}e^{-hk_{p1}}(2\mu_1 p_{11} + \mu_{13}p_{31})$$
$$S_2(6,2) = kk_{p2}e^{-hk_{p2}}(2\mu_1 p_{12} + \mu_{13}p_{32})$$
$$S_2(6,3) = kk_{p3}e^{-hk_{p3}}(2\mu_1 p_{13} + \mu_{13}p_{33})$$
$$S_2(6,4) = -\tfrac{1}{2}e^{-hk_{s1}}\left(k^2 + k_{s1}^2\right)(2\mu_1 q_{11} + \mu_{13}q_{21})$$
$$S_2(6,5) = -\tfrac{1}{2}e^{-hk_{s2}}\left(k^2 + k_{s2}^2\right)(2\mu_1 q_{12} + \mu_{13}q_{22})$$
$$S_2(6,6) = -kk_{p1}(2\mu_1 p_{11} + \mu_{13}p_{31})$$
$$S_2(6,7) = -kk_{p2}(2\mu_1 p_{12} + \mu_{13}p_{32})$$
$$S_2(6,8) = -kk_{p3}(2\mu_1 p_{13} + \mu_{13}p_{33})$$
$$S_2(6,9) = -\tfrac{1}{2}\left(k^2 + k_{s1}^2\right)(2\mu_1 q_{11} + \mu_{13}q_{21})$$
$$S_2(6,10) = -\tfrac{1}{2}\left(k^2 + k_{s2}^2\right)(2\mu_1 q_{12} + \mu_{13}q_{22})$$



$$S_2(7,1) = e^{-hk_{p1}}\left(-(p_{11}S_{c1} + p_{31}S_{c2})k^2 + C_{12}\left(k_{p1}^2 - k^2\right)p_{21} + k_{p1}^2(p_{11}(S_{c1} + 2\mu_1) + p_{31}(S_{c2} + \mu_{13}))\right)$$

$$S_2(7,2) = e^{-hk_{p2}}\left(-(p_{12}S_{c1} + p_{32}S_{c2})k^2 + C_{12}\left(k_{p2}^2 - k^2\right)p_{22} + k_{p2}^2(p_{12}(S_{c1} + 2\mu_1) + p_{32}(S_{c2} + \mu_{13}))\right)$$

$$S_2(7,3) = e^{-hk_{p3}}\left(-(p_{13}S_{c1} + p_{33}S_{c2})k^2 + C_{12}\left(k_{p3}^2 - k^2\right)p_{23} + k_{p3}^2(p_{13}(S_{c1} + 2\mu_1) + p_{33}(S_{c2} + \mu_{13}))\right)$$

$$S_2(7,4) = -e^{-hk_{s1}}kk_{s1}(2q_{11}\mu_1 + q_{21}\mu_{13})$$

$$S_2(7,5) = -e^{-hk_{s2}}kk_{s2}(2q_{12}\mu_1 + q_{22}\mu_{13})$$

$$S_2(7,6) = -(p_{11}S_{c1} + p_{31}S_{c2})k^2 + C_{12}\left(k_{p1}^2 - k^2\right)p_{21} + k_{p1}^2(p_{11}(S_{c1} + 2\mu_1) + p_{31}(S_{c2} + \mu_{13}))$$

$$S_2(7,7) = -(p_{12}S_{c1} + p_{32}S_{c2})k^2 + C_{12}\left(k_{p2}^2 - k^2\right)p_{22} + k_{p2}^2(p_{12}(S_{c1} + 2\mu_1) + p_{32}(S_{c2} + \mu_{13}))$$

$$S_2(7,8) = -(p_{13}S_{c1} + p_{33}S_{c2})k^2 + C_{12}\left(k_{p3}^2 - k^2\right)p_{23} + k_{p3}^2(p_{13}(S_{c1} + 2\mu_1) + p_{33}(S_{c2} + \mu_{13}))$$

$$S_2(7,9) = kk_{s1}(2q_{11}\mu_1 + q_{21}\mu_{13})$$

$$S_2(7,10) = kk_{s2}(2q_{12}\mu_1 + q_{22}\mu_{13})$$

$$S_2(8,1) = e^{-hk_{p1}}(k_{p1} - k)(k + k_{p1})(C_{12}p_{11} + k_2 p_{21} + C_{23}p_{31})$$

$$S_2(8,2) = e^{-hk_{p2}}(k_{p2} - k)(k + k_{p2})(C_{12}p_{12} + k_2 p_{22} + C_{23}p_{32})$$

$$S_2(8,3) = e^{-hk_{p3}}(k_{p3} - k)(k + k_{p3})(C_{12}p_{13} + k_2 p_{23} + C_{23}P_{33})$$

$$S_2(8,4) = 0$$

$$S_2(8,5) = 0$$

$$S_2(8,6) = (k_{p1} - k)(k + k_{p1})(C_{12}p_{11} + k_2 p_{21} + C_{23}p_{31})$$

$$S_2(8,7) = (k_{p2} - k)(k + k_{p2})(C_{12}p_{12} + k_2 p_{22} + C_{23}p_{32})$$

$$S_2(8,8) = (k_{p3} - k)(k + k_{p3})(C_{12}p_{13} + k_2 p_{23} + C_{23}P_{33})$$

$$S_2(8,9) = 0$$

$$S_2(8,10) = 0$$

$$S_2(9,1) = kk_{p1}e^{-hk_{p1}}(\mu_{13}p_{11} + 2\mu_3 p_{31})$$

$$S_2(9,2) = kk_{p2}e^{-hk_{p2}}(\mu_{13}p_{12} + 2\mu_3 p_{32})$$

$$S_2(9,3) = kk_{p3}e^{-hk_{p3}}(\mu_{13}p_{13} + 2\mu_3 p_{33})$$

$$S_2(9,4) = -\frac{1}{2}e^{-hk_{s1}}\left(k^2 + k_{s1}^2\right)(\mu_{13}q_{11} + 2\mu_3 q_{21})$$

$$S_2(9,5) = -\frac{1}{2}e^{-hk_{s2}}\left(k^2 + k_{s2}^2\right)(\mu_{13}q_{12} + 2\mu_3 q_{22})$$

$$S_2(9,6) = -kk_{p1}(\mu_{13}p_{11} + 2\mu_3 p_{31})$$

$$S_2(9,7) = -kk_{p2}(\mu_{13}p_{12} + 2\mu_3 p_{32})$$

$$S_2(9,8) = -kk_{p3}(\mu_{13}p_{13} + 2\mu_3 p_{33})$$

$$S_2(9,9) = -\frac{1}{2}\left(k^2 + k_{s1}^2\right)(\mu_{13}q_{11} + 2\mu_3 q_{21})$$

$$S_2(9,10) = -\frac{1}{2}\left(k^2 + k_{s2}^2\right)(\mu_{13}q_{12} + 2\mu_3 q_{22})$$





$$S_2(10,1) = e^{-hk_{p1}}\left(-(p_{31}S_{c3} + p_{11}S_{c4})k^2 + C_{23}\left(k_{p1}^2 - k^2\right)p_{21} + k_{p1}^2(p_{11}(S_{c4} + \mu_{13}) + p_{31}(S_{c3} + 2\mu_3))\right)$$

$$S_2(10,2) = e^{-hk_{p2}}\left(-(p_{32}S_{c3} + p_{12}S_{c4})k^2 + C_{23}\left(k_{p2}^2 - k^2\right)p_{22} + k_{p2}^2(p_{12}(S_{c4} + \mu_{13}) + p_{32}(S_{c3} + 2\mu_3))\right)$$

$$S_2(10,3) = e^{-hk_{p3}}\left(-(P_{33}S_{c3} + p_{13}S_{c4})k^2 + C_{23}\left(k_{p3}^2 - k^2\right)p_{23} + k_{p3}^2(p_{13}(S_{c4} + \mu_{13}) + p_{33}(S_{c3} + 2\mu_3))\right)$$

$$S_2(10,4) = -e^{-hk_{s1}}kk_{s1}(q_{11}\mu_{13} + 2q_{21}\mu_3)$$

$$S_2(10,5) = -e^{-hk_{s2}}kk_{s2}(q_{12}\mu_{13} + 2q_{22}\mu_3)$$

$$S_2(10,6) = -(p_{31}S_{c3} + p_{11}S_{c4})k^2 + C_{23}\left(k_{p1}^2 - k^2\right)p_{21} + k_{p1}^2(p_{11}(S_{c4} + \mu_{13}) + p_{31}(S_{c3} + 2\mu_3))$$

$$S_2(10,7) = -(p_{32}S_{c3} + p_{12}S_{c4})k^2 + C_{23}\left(k_{p2}^2 - k^2\right)p_{22} + k_{p2}^2(p_{12}(S_{c4} + \mu_{13}) + p_{32}(S_{c3} + 2\mu_3))$$

$$S_2(10,8) = -(P_{33}S_{c3} + p_{13}S_{c4})k^2 + C_{23}\left(k_{p3}^2 - k^2\right)p_{23} + k_{p3}^2(p_{13}(S_{c4} + \mu_{13}) + p_{33}(S_{c3} + 2\mu_3))$$

$$S_2(10,9) = kk_{s1}(q_{11}\mu_{13} + 2q_{21}\mu_3)$$

$$S_2(10,10) = kk_{s2}(q_{12}\mu_{13} + 2q_{22}\mu_3)$$

*Data and code availability.* The data and code that support the findings of this study can be found in (Hongwei et al., 2021) or https://github.com/Siglab-code/WaveFrost.

*Author contributions.* Conceptualization: Hongwei Liu, Pooneh Maghoul; Methodology: Hongwei Liu, Pooneh Maghoul; Investigation: Hongwei Liu, Pooneh Maghoul; Visualization: Hongwei Liu, Pooneh Maghoul; Supervision: Pooneh Maghoul, Ahmed Shalaby; Writing—original draft: Hongwei Liu; Writing—review & editing: Pooneh Maghoul, Ahmed Shalaby

*Competing interests.* Pooneh Maghoul has patent Systems and Methods for In-situ Characterization of Permafrost Sites pending to Pooneh Maghoul; Hongwei Liu; Guillaume Mantelet; Ahmed Shalaby; University of Manitoba.

*Acknowledgements.* The authors would like to acknowledge National Science Centre, Poland (NCN) UMO-2016/21/B/ST10/02509 for the support of the MASW permafrost measurements. The authors are grateful to Dr. Mariusz Majdański, Mr. Artur Marciniak and Mr. Bartosz Owoc for sharing the data. The authors also acknowledge the financial support of the New Frontiers in Research Fund - Exploration Grant [NFRF-2018-00966], the Natural Sciences and Engineering Research Council of Canada (NSERC) - Discovery Grant program [RGPIN-2016-06019], the Mathematics of Information Technology and Complex Systems (Mitacs) Accelerate program, and the University of Manitoba Graduate Enhancement of Tri-Council Stipends (GETS) program.





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
