# Peer review of "Seismic physics-based characterization of permafrost sites using surface waves"

_The Cryosphere, 2021_

## Referee Comment (RC2)

**General comments**

The main contribution of this article is the identification of two wave modes that interact sensitively with the porous and mechanical properties of a layered poro-elastic medium, respectively. In addition the authors use a multiphase poro-mechanical foundation to build a global matrix model of a layered poro-elastic medium. This approach has significant novelty compared to the typical global matrix models based on partial wave amplitudes and nicely emphasizes the connection between surface waves and ground physical properties. The specific application to permafrost demonstrates the relevance of this manuscript to The Cryosphere, although certain aspects of the methodology also have broader relevance to the field of near surface geophysics and non-destructive testing of engineering materials.

The weaker side of the manuscript is that findings that are a direct result of the data examples presented are not adequately separated from other applications that remain essentially hypothetical (these are detailed under "specific comments"). The manuscript could also be improved substantially by giving a more complete description of the two Rayleigh wave modes so that those reading the manuscript may better understand their propagation and interpret their broader relevance to the field of surface wave seismic investigation. Furthermore, I have some concerns that the inversion results are overly sensitive to the frequency range of the dispersion curves that constrain the inversion (likely due to a mismatch between the shape of the experimental and inverted dispersion curves). The anomalous result at 360-480 m in the physical data example is not convincing and appears more likely to reflect a weakness in the inversion methodology than real lateral variation in physical properties. Furthermore, it is generally difficult to assess the true accuracy of the results owing to a lack of comparison to ground truth observations of physical properties and interface depths or comparison with other geophysical datasets (both of which appear to exist in the published literature). I believe it should be possible for the authors to address these concerns in a revised manuscript, that will then make a useful contribution to The Cryosphere.

**Specific comments**

1. The authors claim that their methodology can be used to "characterize a permafrost site more accurately" (line 12). However, since we have no baseline for comparison it is difficult to assess to what extent this is the case. The Glazer (2020) study that is already cited by the authors could be used as a direct comparison in order to place the results of the present study in context. The ERT results of Glazer (2020) may provide a means of independent validation, while the MASW results of Glazer (2020) use more conventional processing of the same dataset as used in this study and could provide an excellent benchmark to highlight the benefits of the new hybrid inversion approach.

   Szymański et al. (2013) have published direct sampling results of 34 soil pits for the Fuglebekken area. It would be highly valuable for the authors to refer to this study in order to place their results in a geological context. Significantly, Szymański et al. (2013) describe the area as consisting of crystalline bedrock covered by marine deposits with thickness up to 4-5 m. Is there a possibility that the interface between the active layer and the ice bearing permafrost that the authors place at ~4 m depth is really the sediment-bedrock interface?

Szymański, W., Skiba, S., & Wojtuń, B. (2013). Distribution, genesis, and properties of Arctic soils: a case study from the Fuglebekken catchment, Spitsbergen. *Polish Polar Research*, 289-304.

2. The dispersion spectra for the R2 wave (Fig. 1b) looks very similar to the Rayleigh-Lamb waves described by Ryden & Park (2004) for pavements and also shown to occur in permafrost settings by Romeyn et al. (2021), although the experimental data in the present study does not resolve the higher order modes. It would be beneficial for the authors to refer to this work, particularly since the global matrix method employed by Ryden & Park (2004) is similar to the theoretical development of this manuscript. It may be purely an issue of terminology, but the authors could also consider that the surface waves identified in the manuscript may be more accurately considered Rayleigh-Lamb waves, since the stiff ice-bearing permafrost layer likely acts as a waveguide to some extent. The following passage from Ryden & Park (2004) provides some perspective on this topic:

> *"It is usually assumed that Rayleigh waves are the prevailing type of waves generated, with a depth penetration of about one wavelength (Viktorov 1967). However, it has been reported that this assumption holds strictly only at sites where the stiffness increases smoothly as a function of depth (Foti 2000). At sites with a velocity reversal (i.e. stiffness decreases with depth), the nature of surface-wave propagation has been reported as more complicated than at sites with normal dispersion. Several studies have indicated that a measured dispersion curve where the phase velocity increases with frequency, i.e. inverse dispersion, is actually built up by small portions of higher modes (Gucunski and Woods 1992; Tokimatzu et al. 1992; Forbriger 2003; Foti et al. 2003; Ryden et al. 2004)."*

Rydén, N., & Park, C. B. (2004). Surface waves in inversely dispersive media. *Near surface geophysics*, *2*(4), 187-197.

Romeyn, R., Hanssen, A., Ruud, B. O., Stemland, H. M., & Johansen, T. A. (2021). Passive seismic recording of cryoseisms in Adventdalen, Svalbard. *The Cryosphere*, *15*(1), 283-302.

3. The proposed application in "early detection and warning systems to monitor infrastructure impacted by permafrost-related geohazards" (line 12) is not sufficiently developed. It should be established by data, reference to other studies or at least step-by-step logic that precursor change in physical properties could be detected by the proposed monitoring methodology in advance of changes that result in structural damage, for example. If such evidence does not exist, this application should be limited to a briefly describing that this early detection system is a goal that will be pursued in future studies.

4. The possibility to "detect the presence of layers vulnerable to permafrost carbon feedback and emission of greenhouse gases into the atmosphere" (line 13) is not convincingly demonstrated and may overemphasize the direct relevance of this study to assessment of the global carbon budget. If one reads carefully through the manuscript, this statement comes down to the hypothesized ability of the study methodology to detect the presence of peat in the subsurface. The authors have not argued how distinct the physical parameters of peat are from other soil

types and to what level of confidence it could be detected in practice. Ideally, there should be at least one real data example of a known peat layer being detected by proposed methodology. Synthetic data could also be usefully employed to demonstrate the hypothesized application. I would further suggest that the authors describe specifically that there are two steps, 1) detection of peat layers 2) estimation of carbon content. It should otherwise be clearly demonstrated that the variation of organic carbon content of soils that are otherwise similar leads to a detectible variation in mechanical properties using Rayleigh wave modes 1 & 2.

5. Line 16-19. Missing reference. In particular "The thickness of the active layer depends on local geological and climate conditions such as vegetation, soil composition, air temperature, solar radiation and wind speed" should be supported with one or more references. The active layer undergoes seasonal freeze and thaw cycles by definition so to say "may undergo" seems strange (I assume this was an oversight and have added a technical correction).

6. Line 24. Missing reference. Please add one or more references that support the excessive deformation in frost-susceptible soils caused by segregated ice formation.

7. Line 27. Missing reference. Please add one or more references that describe the ice-wedge formation process and its timescale.

8. Line 33. Missing reference. Please add one or more references that describe thaw settlement and associated loss of strength.

9. Line 56. Missing reference. It is stated that it is common practice to associate anomalously high shear wave velocities with permafrost, but no references are given to studies that are examples of this practice.

10. Line 64. Similar to point 3. It is not sufficiently clear what the authors mean by "we can also predict the soil type and the sensitivity of the permafrost layer to permafrost carbon feedback and emission of greenhouse gases to the atmosphere". It seems natural to guess that this means the ability to quantify the amount of organic carbon present in the soil, but it is not clear to what extent the proposed methodology is capable of this.

11. Figure 2. The layer stiffness matrices should be defined in the caption. Perhaps this figure should be dropped entirely since it is minimally illustrative when the layer matrices are only given in the appendices.

12. Line 108. "The global stiffness matrix for the R1 wave can be decomposed into the components related only to the P1 and S1 wave velocities." There should be a reference to an equation associated with this statement.

13. Line 109. "proved that the R1 wave is generated by the interaction between the P1 and S1 waves." Include a reference to section 3 where this argument is made so that the reader does not get lost here.

14. Line 151. "The seismic measurements shown in Figure 3a are indeed a combination of both R1 and R2 waves." These are not seismic measurements; they are synthetic data. It would perhaps be better to say "The surface waves shown in…". Is the conclusion that Fig. 3a shows R1 and R2 waves based on the velocity match? It could be clearer what the authors are trying to convey with this sentence.

15. Figure 3. There is a lot of wasted space with zero amplitudes in Fig. 3a. While it shows the velocity moveout, it fails to illustrate the detailed waveforms of the R1 and R2 waves. It would be very useful to include at least one detailed timeseries example to illustrate the waveforms of these waves. The identification of the R1 and R2 waves is a key part of the main contribution of this article and I do not think the current figure is adequate to illustrate their characteristics. There is also no colour scale on Fig. 3b-c.

16. Line 176. How are the experimental dispersion spectra obtained? One assumes the phase shift method of Park (1999), but it is not stated. Are the dispersion curves manually picked or fitted to the spectral peak by a semi-automatic method? The only detail given is "The R1 and R2 Rayleigh waves are identified by visual inspection to obtain the experimental dispersion relations". This seems radically insufficient given that the experimental dispersion relations are the key inversion constraint. One would expect that every detail surrounding the dispersion curve picking should be fully accounted for given their importance to the manuscript.

Park, C.B., Miller, R.D., and Xia, J., 1999, Multichannel analysis of surface waves (MASW): Geophysics, 64, 800–808.

17. Line 185. "The unfrozen ground is believed to have a degree of saturation of unfrozen water of about 100% (fully saturated)" is this based on previous studies? Please give a reference to support the assumption.

18. Line 194: It does not seem clear that the soil type, it's mineral composition or it's organic carbon content have been resolved in the present study. It seems defensible that the presence of an ice rich layer has been demonstrated but the authors seem to be speculating far beyond this and these speculations should be either moderated or backed up with real or synthetic data examples to illustrate that they are feasible.

19. Line 201: "Given the high ice-to-water ratio, we therefore interpret the permafrost is currently in a stable frozen state." The logic is unclear or not fully developed here. Unstable permafrost is distinguished by whether the ice and water saturation exceed the total pore volume of the ground in an unfrozen state. It seems that 22% water plus 91% ice (maximum values of uncertainty ranges) could exceed the pore volume in the permafrost layer and could therefore be considered unstable, though on the lower end 8.8% water plus 77% ice would be less than

the pore volume in the frozen state of the permafrost. However, the frozen permafrost has elevated porosity compared to both the overlying and underlying layers. If we consider that water plus ice saturation in the permafrost is at least 86% of a frozen pore volume of 0.43-0.46, then it will likely be more than 100% saturation if the porosity in the unfrozen state is approximated by the porosity of the overlying active layer and underlying unfrozen ground (represented as 0.34-0.36 if we take the zone of overlap between the two porosity estimates). I think the authors should explicitly step through their assumptions and calculations in determining the stability of the permafrost, because this is a significant application area of their methodology and could be a major strength of the work if developed to its full potential.

20. Figure 4. The dispersion spectra use a colour scale that has an orange colour which appears in two different amplitude ranges making interpretation of the spectra ambiguous. I also don't understand why the manually picked dispersion curves are not overlain on the spectra. This is particularly important because the spectra are somewhat poorly resolved and identification of the precise dispersion relation is far from straightforward from these data.

21. Figure 5. It is concerning that the prediction envelope of the R2 dispersion relation is concave down at frequencies below 20 Hz, while the experimental measurement is concave up. Looking at the experimental dispersion image for the R2 wave in Fig. 4d it seems quite clear that low frequencies should trend towards high phase velocities. The implication here is that some important parameter of the system is not resolved by the inversion. I will come back to this, but it may be the root cause of the anomalous result reported for the 360-480m section.

22. Line 225. "the permafrost table is generally located at about 4 m below the ground surface, except at the offset distance from 360 m to 480 m where the permafrost table is located at 1.1 m below the ground surface." Is there a geologic or geomorphologic explanation for this variation, e.g., topography, vegetation, surface-water etc.? Is there otherwise some other geophysical dataset that could corroborate this? It seems rather implausible that the permafrost table is so dramatically elevated at one anomalous location. If the anomalous result at 360-480 m cannot be explained from a reasonable physical or geological perspective then it rather points towards a significant degree of uncertainty or instability in the inversion.

23. Line 233. "Sufficient agreement exists between the numerical and experimental dispersion relations for the R2 wave (Figure 7d) which confirms the acceptance of the predicted values for the volumetric ice content (calculated as the product of porosity and the degree of saturation of ice) and porosity ". I find it difficult to agree with this statement. The model and experimental dispersion curves have notably poorer correspondence for the 360-480 m section, which is the only section that gives a significantly different inversion result. This points towards model misfit rather than physical reality.

24. Figure 7. It is not convincing that the anomalously shallow permafrost table, high ice content result at 360-480 m reflects a real variation in ground structure/properties. The experimental dispersion curves look quite similar in the overlapping frequency ranges, but the 360-480 m dispersion curve extends to lower frequencies than the others do. It would be beneficial to

examine a figure plotting all dispersion curves on a shared axis so the reader can see where and by how much they really vary (but this is of course up to the authors discretion). In all cases, it looks like the experimental dispersion curves are concave up at low frequencies and the R2 prediction envelopes are concave down. This mismatch is exacerbated for the 360-480 m section, which extends to lower frequencies than the others and therefore leads to the anomalous result. It is difficult to say which result is closer to reality because of a lack of comparison with ground truth observations or other geophysical data sets. The frequency range from ~13-20 Hz is where the phase velocity of the R2 wave varies most significantly (Fig 4d) so it is concerning that the inversion seems to have problems matching the experimental curve in exactly this part of the frequency spectrum.

25. Line 238. "at the offset distance from 360 m to 480 m the coldest temperature of about -12 ∘C (Figure 7e) occurs in the permafrost layer, which is highly related to the high ice content in this section." Again, a more nuanced interpretation is required. It is difficult to accept that the anomalous data section, with the poorest correspondence between model and experimental dispersion curves can simply be interpreted as a real physical effect without giving a supporting physical explanation.

26. Line 253. "the mechanical properties of the solid skeletal frame can reveal the type of soil". How much overlap in mechanical properties is there for different types of soils and how does the estimation compare with the soil pit sampling study of, e.g., Szymański et al. (2013) which covers the Fuglebekken area?

27. Line 256. "if the mechanical properties of the solid skeletal frame correspond to the ones for peat we can perform more detailed investigation to assess the sensitivity of the permafrost to greenhouse gases emission." It is important to communicate that this application remains hypothetical, since the ability to resolve the presence of a peat layer has not been demonstrated in this study. Perhaps the authors would consider adding a synthetic data example including a peat layer if they feel this is an important application to emphasize.

28. Line 260. "we can reasonably consider the permafrost layer at the offset distance from 360 m to 480 m to be ice-rich and ice segregation layers are expected to contribute to its relatively higher volumetric ice content." This seems to require an assumption of the porosity in the unfrozen state, which is not given explicitly but should be, so that the reader can follow the authors line of reasoning. It would also be valuable to discuss if it is physically reasonable for a change to occur at this location alone, while all other locations consistently gave a different result.

29. Line 267. "The uncertainty originates from the non-uniqueness in the inverse analysis (local minima problem) and the limited number of constraints in the inversion analysis". The sensitivity to small changes in the experimental dispersion curves is not adequately covered in the manuscript. For example, the 360-480 m section has an experimental dispersion curve that appears quite similar to the other sections, but extends to a lower frequency range and gives a substantially different inversion result. More generally, there is always some uncertainty in picking the dispersion curve from experimental data and it is unclear how this uncertainty may

propagate through the inversion. How do the results differ for a set of dispersion curves that are indistinguishably close from an experimental perspective? The R1 dispersion spectra in particular is quite poorly resolved (Fig. 4c) so one must assume some degree of uncertainty is associated with the picked dispersion curve.

30. Line 268. "recommended to use other geophysical methods to improve the resolution and reduce uncertainty of the permafrost mapping." Why are the inversion results of the field example not discussed in the context of existing geophysical and direct sampling results? This is a crucial step in qualifying the validity of the proposed methodology.

31. Line 272. "The proposed hybrid inverse and multi-phase poro-mechanical approach can potentially be used for the design of an early warning system for permafrost by means of an active or passive seismic test." It seems that too much emphasis is placed on this hypothetical future application while the more important topic of qualifying the inversion results in the context of other geophysical methods, direct sampling, geological and geomorphological understanding etc. is lacking. There is no convincing argument that changes in poro-mechanical properties that would be detectable with the current methodology occur in advance of physical surface expressions such as subsidence or cracks in structures. This would presumably be a key requirement of an early warning system.

32. Line 277. "The early warning system can provide long-term tracking of permafrost conditions particularly when the ice content or mechanical properties of permafrost approach critical values." What are the critical values? Again, either the concept of an early warning system should be developed fully and convincingly, or it should just be mentioned briefly as a goal for future research efforts.

**Technical corrections**

1. It would be much easier to read if references to appendices were presented in the form "Appendix D" not simply "D" e.g., line 90 "the matrix… are given in D" would become ""the matrices… are given in Appendix D""
2. Line 17 "the active layer, may undergo seasonal thaw and freeze cycles" should be "the active layer, undergoes seasonal thaw and freeze cycles"
3. Line 29 "This distinction is determined by the amount of ice content within the permafrost." Should be "This distinction is determined by the ice content within the permafrost." OR "This distinction is determined by the amount of ice within the permafrost.", amount and content both refer to the same quantity here.
4. Line 30 "Ice-rich permafrost contains ice in excess of its water content at saturation." Could be modified to "Ice-rich permafrost contains ice in excess of its water content at saturation and is thaw unstable." In order to improve the flow of argumentation in the surrounding paragraph.
5. Line 47 "GPR has been also used" should be "GPR has also been used"
6. Line 50. "none of the above-mentioned methods characterizes the mechanical properties of permafrost layers." Should rather be "none of the above-mentioned methods directly characterizes the mechanical properties of the permafrost."

7. Line 67. "based on an MASW seismic investigation in a field located at SW Spitsbergen, Norway" should rather be "based on a MASW seismic investigation of a field site located on SW Spitsbergen, Svalbard".

8. Line 77. "A random sample is initially generated to ensure that soil parameters are not affected by a local minimum" makes it sound as if it is a single initial sample. I think the following might be a more correct representation of what the authors mean to express "A set of initial values, randomly selected and spanning the multidimensional parameter space ensures that soil parameters are not affected by a local minimum". Same comment applies to line 123.

9. Figure 1 caption. "Dispersion relations of R1 and R2 waves" should be "Dispersion image of R1 and R2 waves". The annotation on figure panel (b) should also be changed since the figure shows dispersion images and not curves.

10. Line 167. It is more geographically descriptive to write SW Spitsbergen, Svalbard (rather than Norway).

11. Line 169. Why not give the number of geophones directly? E.g. "The MASW test was performed by using 60 geophone receivers spaced at regular 2m intervals".

12. Line 193. "detection of the thin ice lenses using low frequency seismic waves is highly impossible due to the mismatch between the thickness of the ice segregation layers and the wavelength generated in seismic tests". It is not valid to say "highly impossible". Why not simply say that ice lenses cannot be detected directly below 1/4 lambda, or whatever fraction of a wavelength is believed to be the correct detection limit here? To describe the phenomenon as a mismatch between wavelength and thickness is rather vague.

13. Line 201. "with a nearly 8.8%-22% of degree of saturation" it does not make sense to say nearly followed by a range, just give the range and omit "nearly".

14. Line 209. "sufficiently close" is a highly subjective description. "show good visual agreement" is perhaps what the authors intend to convey, but the phrasing should be made more descriptive in any case.

15. Figures 5, 6, B1-B4 and line 258 in text "Saturation degree" should be "degree of saturation" which is the correct terminology and that which is mostly used throughout the text.

---

## Author Response (AR1)

**Subject:** Detailed Responses to Reviewers: Liu, H., Maghoul, P., and Shalaby, A.: Seismic physics-based characterization of permafrost sites using surface waves, The Cryosphere Discuss. [preprint], `https://doi.org/10.5194/tc-2021-219`, in review, 2021.

**Date:** January 12, 2022
* * *
The authors are grateful for the valuable comments and kind consideration of our submission. Detailed responses and revisions based on these comments are listed below.

Reviewer 1

**General Comments**

This article proposes an original use of seismic methods to characterize a permafrost area. The main interest of the study lies in the identification and interpretation of two types of Rayleigh waves propagating in a frozen porous medium. The separate inversion of the two dispersion curves provides an hybrid method for determining independently the physical and mechanical properties of the medium, thanks to the difference in the respective sensitivity of these two waves to these properties. The article invites the use of this method to characterize a permafrost medium, as it appears to be more efficient and requires fewer a priori assumptions about the investigated medium.

The authors mention various applications to the detection and characterization of permafrost, ranging from civil engineering and infrastructure monitoring to the assessment of the potential vulnerability of certain areas to permafrost degradation and associated feedbacks.

The article is well structured and adequately written. A significant contribution is that authors used seismic data collected at a site in Svalbard, and applied their processing to this experiment, to show a real application of their method.

In my opinion the paper deserves publication after minor revisions.

**1.** First, the contribution of this study to the current knowledge of seismic waves propagating in permafrost is not very comprehensible to the reader. The lack of references about the poroelastic model and the lack of physical interpretation of the two Rayleigh waves should be corrected.

From a poromechanical point of view, permafrost (frozen soil) is a multi-phase porous medium that is composed of a solid skeletal frame and pores filled with water and ice with different proportions. Three types of P wave (P1, P2 and P3) and two types of S wave (S1, S2) coexist in three-phase frozen porous media (Carcione et al., 2000; Carcione and Seriani, 2001; Carcione et al., 2003). The P1 and S1 waves are the longitudinal and transverse waves propagating in the solid skeletal frame, respectively, but are also dependent on the interactions with pore ice and pore water (Carcione and Seriani, 2001). The P2 and S2 waves propagate mainly within pore ice (Leclaire et al., 1994). Similarly, the P3 wave is due to the interaction between the pore water and the solid skeletal frame. However, the understanding of surface wave propagation in permafrost is still limited in the literature. The current surface wave analysis in foundation permafrost does not consider the interaction of different wave modes (P1, P2, P3, S1 and S2) due to the multiphase poroelastic properties of permafrost at the near surface and still assume permafrost soils as a solid elastic material (Leblanc et al., 2006; 2017; Krautblatter et al., 2014, Dou et al., 2014; Ajo-Franklin et al.). In this paper, we have identified and demonstrated the formation of two types of Rayleigh waves (R1 and R2) at the surface in permafrost sites due to the interaction of different phases (e.g., solid skeletal frame, pore-water and pore-ice). More importantly, we concluded that the phase velocity of the R1 wave is mostly sensitive to the shear modulus of the solid skeletal frame; it is also dependent on the bulk modulus, porosity, and degree of saturation of ice. On the other hand, the phase velocity of the R2 wave is almost independent of the mechanical properties of the solid skeletal frame, while it is strongly affected by the porosity and degree of saturation of ice, as shown in Figure 1e and 1f (or Figure 2 in the revised manuscript). The detailed discussion is also given in Section 3 from line 131 to 176 in the revised manuscript.

References about the three-phase poroelastic model used in this paper have been added to the revised manuscript line 305-306 (Leclaire et al., 1994 and Carcione et al., 2000).

To physically interpret the two Rayleigh waves, a uniform frozen soil layer is used to show the propagation of different types of P and S waves and subsequently the formation of Rayleigh waves (R1 and R2) at the surface. It is assumed that an impulse load with a dominant frequency of 100 Hz is applied at the ground surface. The wave propagation analysis was performed in clayey soils by assuming a porosity (n) of 0.5, a degree of saturation of unfrozen water (Sr) of 50%, a bulk modulus (K) of 20.9 GPa and a shear modulus (G) of 6.85 GPa for the solid skeletal frame (helgerud et al., 1999). The velocities of the P1 and P2 waves are calculated as 2,628 m/s and 910 m/s, respectively, based on the relations given in Appendix A in the manuscript. The velocity of P3 wave (16 m/s) is relatively insignificant in comparison to P1 and P2 wave velocities. Similarly, the velocities of the S1 and S2 waves are calculated as 1,217 m/s and 481 m/s, respectively. Accordingly, the observed displacements measured at the ground surface with an offset from the impulse load ranging from 0 to 120 m are illustrated in Figure 1a. Figure 1b to illustrate the waveforms of R1 and R2 waves at the offset of 80 m. Figure 1c and 1d illustrate the appearance of two types of Rayleigh waves (R1 and R2) in a three-phase permafrost subsurface at 70 ms and 100 ms, respectively. We found the velocity of R1 and R2 is 1,150 m/s and 450 m/s using the three-phase dispersion relation derived in this paper. It is commonly known that the Rayleigh wave is slightly slower than the shear wave velocity and the ratio of Rayleigh wave and shear wave velocity ranges from 0.92-0.95 for Poisson's ratio greater than 0.3 (Kazemirad et al., 2013). From this analysis, we found the ratio of R1 and S1 wave velocity is around 0.93. Similarly, the ratio of R2 and S2 wave velocity is around 0.94. Therefore, we can conclude that R1 waves appear due to the interaction of P1 and S1 waves since the phase velocity of R1 waves is slightly slower than the phase velocity of S1 waves. Similarly, R2 waves appear due to the interaction of P2 and S2 waves since the phase velocity of R2 waves is also slightly slower than the phase velocity of S2 waves. The detailed discussion is also given from line 141 to 162 in the revised manuscript.

[Figure]

Figure 1: **(a)** Theoretical time-series measurements for R1 and R2 Rayleigh waves at the ground surface **(b)** Waveforms of R1, R2 and other wave modes at the offset of 80 m. **(c)** Displacement contour at time 70 ms. **(d)** Displacement contour at time 100 ms with the labeled R1 and R2 Rayleigh waves. **(e)** Effect of shear modulus and bulk modulus of the solid skeletal frame on phase velocity of R1 and R2 waves. **(f)** Effect of degree of saturation of ice on the phase velocity of R1 and R2 waves.

Refernce:

Ajo-Franklin, J., Dou, S., Daley, T., Freifeld, B., Robertson, M., Ulrich, C., & Wagner, A. (2017). Time-lapse surface wave monitoring of permafrost thaw using distributed acoustic sensing and a permanent automated seismic source. In SEG Technical Program Expanded Abstracts 2017 (pp. 5223-5227). Society of Exploration Geophysicists.

Leblanc, A. M., Fortier, R., Cosma, C., & Allard, M. (2006). Tomographic imaging of permafrost using three-component seismic cone-penetration test. Geophysics, 71(5), H55-H65.

Krautblatter, M., & Draebing, D. (2014). Pseudo 3-D P wave refraction seismic monitoring of permafrost in steep unstable bedrock. Journal of Geophysical Research: Earth Surface, 119(2), 287-299.

[revised manuscript text omitted]

The main contribution of this study is that we proposed a hybrid inverse and multi-phase poromechanical approach for in-situ characterization of permafrost sites using the decomposition of two Rayleigh waves. In this method, we quantify the physical properties such as ice content, unfrozen water content, and porosity as well as the mechanical properties such as the shear modulus and bulk modulus of permafrost or soil layers. The MASW seismic investigation in the field site located at SW Spitsbergen, Svalbard is used to demonstrate the role of two different types of Rayleigh waves in characterizing the permafrost. Our results demonstrate the potential of seismic surface wave testing accompanied by our proposed hybrid inverse and poromechanical dispersion model for the assessment and quantitative characterization of permafrost sites. The detailed discussion is also given in line 62-72 in the revised manuscript. The highlights of this research include:

- Proposed a novel physics-based signal processing algorithm to quantitatively estimate the physical and mechanical properties of a permafrost site by surface waves

- Identified the existence of two types of Rayleigh waves (R1 and R2) where R1 travels relatively faster than R2 in a permafrost site

- The R1 wave velocity depends strongly on the soil type and mechanical properties (e.g., shear modulus and bulk modulus) of permafrost layers

- The R2 wave velocity is highly sensitive to the physical properties (e.g., unfrozen water content, ice content) of permafrost layers

Reference:

Glazer, M., Dobiński, W., Marciniak, A., Majdański, M., & Błaszczyk, M. (2020). Spatial distribution and controls of permafrost development in non-glacial Arctic catchment over the Holocene,

Fuglebekken, SW Spitsbergen. Geomorphology, 358, 107128.

Szymański, W., Skiba, S., & Wojtuń, B. (2013). Distribution, genesis, and properties of Arctic soils: a case study from the Fuglebekken catchment, Spitsbergen. Polish Polar Research, 289-304.

Olafsdottir, E. A., Erlingsson, S., & Bessason, B. (2018). Tool for analysis of multichannel analysis of surface waves (MASW) field data and evaluation of shear wave velocity profiles of soils. Canadian Geotechnical Journal, 55(2), 217-233.

[Figure]

Figure 2: Surface wave measurement in Section 1 (from 0 m to 120 m). **(a)** Study area in Holocene, Fuglebekken, SW Spitsbergen. **(b)** Test site with clayey silt soils. **(c)** Test site with gravels and sands. **(d)** Test site with patterned ground. **(e)** Waveform data from the measurements at different offsets in horizontal distance. **(f)** Experimental dispersion image for R1 wave. **(g)** Experimental dispersion image for R2 wave

**3.** More generally, there is a lack of references addressing issues which the authors mention. For an example, the applications (early warning systems and permafrost carbon feedback vulnerability) are frequently mentioned, but have to be more documented.

In this paper, our results demonstrate the potential of seismic surface wave testing accompanied with our proposed hybrid inverse and poromechanical dispersion model for the assessment and quantitative characterization of permafrost sites. Its applications for early detection and warning systems to monitor infrastructure impacted by permafrost-related geohazards, and to detect the presence of layers vulnerable to permafrost carbon feedback and emission of greenhouse gases into the atmosphere will be the goal of our future studies. Currently, there is no advanced physics-based monitoring system developed for the real-time interpretation of seismic measurements. As such, active and passive seismic measurements can be collected and processed using the proposed hybrid inverse and poromechanical dispersion model for the assessment and quantitative characterization of permafrost sites at various depths in real-time. In the future study, we will focus on the development of an early warning system for the long-term tracking of permafrost conditions. The early warning system can be used to collect seismic measurements and predict the physical and mechanical properties of the foundation permafrost. The system then reports periodic variations in physical (mostly ice content) and mechanical properties of the permafrost being monitored. The same method being applied on different dates (e.g. seasonal basis) can be used to record the change of properties of the permafrost site, and then warn on the degradation of the permafrost exceeding the threshold. The value of the threshold (or critical values) will require more in-depth research to be determined. The early detection and warning systems can be beneficial in monitoring the condition of the foundation permafrost and preventing excessive thawing settlement and significant loss in strength. Similarly, we can detect the presence of peat (based on the physical and mechanical properties) which is vulnerable to permafrost carbon feedback and emission of greenhouse gases into the atmosphere. It's reported that the soils in the permafrost region hold twice as much carbon as the atmosphere does (almost 1,600 billion tonnes) (Schuur et al., 2015). The thawing permafrost can rapidly trigger landslides and erosion. Current climate models assume that permafrost thaws gradually from the surface downwards (Schuur et al., 2015). However, several meters of soil can become destabilized within a few days or weeks instead of a few centimeters of soil thawing each year (Schuur et al., 2015). The missing element of the existing studies and models is that the abrupt permafrost destabilization can occur and contribute to more carbon feedback than the existing models predict as the permafrost degrades. The detailed discussion is also given in line 287 to 309 in the revised manuscript.

Reference:

Schuur, E. A., McGuire, A. D., Schädel, C., Grosse, G., Harden, J. W., Hayes, D. J., & Vonk, J. E. (2015). Climate change and the permafrost carbon feedback. Nature, 520(7546), 171-179.

**4.** Finally, uncertainties of this new method must be addressed more quantitatively, in order to better assess its benefits and drawbacks over other methods.

Root mean square value (RMS) has been added in the manuscript to quantify the misfit between the experimental and numerical dispersion curves for both R2 and R1 waves, as shown in Figure 3 (or Figure 4 in the revised manuscript).

The uncertainties due to the selection of the dispersion curve from the dispersion spectra have been considered in the revised manuscript. The dispersion curve is automatically selected initially based on the highest intensity in the dispersion spectra using MASWave software (Olafsdottir et al., 2018). Then a 90% confidence interval (labeled as lower bound, highest intensity and upper bound, as shown in Figure 2f and 2g) is considered to study the uncertainties of the selection of dispersion curve to the inversion results. Finally, a range for each parameter (e.g., the degree of saturation of unfrozen water, porosity, shear modulus and bulk modulus) is given to quantify the uncertainty. The detailed discussion is also given in line 199 to 203 in the revised manuscript.

For instance, Figure 3a shows the probabilistic distribution of the degree of saturation of unfrozen water with depth in Section 1. Our results show that the active layer has a thickness of about 1.5 m. The predicted permafrost layer (second layer) has a nearly 32% of degree of saturation of unfrozen pore water. Figure 3b shows the degree of saturation of ice with depth. The degree of saturation of ice in the permafrost layer (second layer) ranges from 67% to 71%. Figure 3c illustrates the porosity distribution with depth. The porosity is around 0.60 in the first layer (active layer), from 0.40 to 0.47 in the second layer (permafrost) and from 0.56 to 0.59 in the third layer. Figure 3d and 3e show the predicted mechanical properties of the solid skeletal frame (shear modulus and bulk modulus) in each layer. It was reported by Szymański et al. (2013) that this study site also contains a lot of coarse sandy soils, gravels as well as around 20% silty clay based on the direct sampling methods at the top 15 cm. The predicted shear modulus and bulk modulus for the solid skeletal frame in the permafrost layer (second layer) are about 13 GPa and 12.7 GPa, which are in the range for silty-clayey soils (Vanorio et al. 2003) and are also consistent with the local soil types described by Szymański et al. (2013). The predicted shear modulus and bulk modulus for the solid skeletal frame in the third layer are about 4 GPa and 10 GPa, which are in the range for clayey soils (Vanorio et al. 2003). Figure 3f and 3g show the comparison between the numerical and experimental dispersion relations for R2 and R1 waves, respectively. The numerical predictions show good agreement with the experimental dispersion curves for both R1 and R2 waves. The detailed discussion is also added in line 219 to 232 in the revised manuscript.

Reference:

Olafsdottir, E. A., Erlingsson, S., & Bessason, B. (2018). Tool for analysis of multichannel analysis of surface waves (MASW) field data and evaluation of shear wave velocity profiles of soils. Canadian Geotechnical Journal, 55(2), 217-233.

Vanorio, T., Prasad, M., & Nur, A. (2003). Elastic properties of dry clay mineral aggregates, suspensions and sandstones. Geophysical Journal International, 155(1), 319-326.

Szymański, W., Skiba, S., & Wojtuń, B. (2013). Distribution, genesis, and properties of Arctic soils: a case study from the Fuglebekken catchment, Spitsbergen. Polish Polar Research, 289-304.

[Figure]

Figure 3: Surface wave inversion results for Section 1: 0m to 120m. **(a)** Degree of saturation of unfrozen water, **(b)** Degree of saturation of ice, **(c)** Porosity distribution, **(d)** Shear modulus of solid skeletal frame, **(e)** Bulk modulus of solid skeletal frame, **(f)** Experimental and numerical dispersion curves for R2 wave, **(g)** Experimental and numerical dispersion curves for R1 wave.

**Specific comments**

Applications : early warning systems and permafrost carbon feedback vulnerability: I suggest to add more details about what could be applied, and more referenced. Otherwise, these applications would be mention with caution only in the discussion part.

Its applications for early detection and warning systems to monitor infrastructure impacted by permafrost-related geohazards, and to detect the presence of layers vulnerable to permafrost carbon feedback and emission of greenhouse gases into the atmosphere are provided with more details. Active and passive seismic measurements can be collected and processed using the proposed hybrid inverse and poromechanical dispersion model for the assessment and quantitative characterization of permafrost sites at various depths. In the future study, we will focus on the development of an early warning system for the long-term tracking of permafrost conditions. The early warning system can be used to collect seismic measurements and predict the physical and mechanical properties of foundation permafrost. The system then reports a periodic variation of physical (mostly ice content) and mechanical properties of the permafrost being monitored over time. The same method being applied on different times (e.g. seasonal basis) can be used to record the change of properties of the foundation permafrost, and then warn on the level of degradation of the permafrost exceeding the threshold. The value of the threshold (or critical values) will require more in-depth research to be determined. The early detection and warning systems can be beneficial in monitoring the conditions of the foundation permafrost and preventing excessive thawing settlement and significant loss in strength. Similarly, we can detect the presence of peat (based on the physical and mechanical properties) which is vulnerable to permafrost carbon feedback and emission of greenhouse gases into the atmosphere. It's reported the soils in the permafrost region hold twice as much carbon as the atmosphere does (almost 1,600 billion tonnes) (Schuur et al., 2015). The thawing permafrost can rapidly trigger landslides and erosion. Current climate models assume that permafrost thaws gradually from the surface downwards (Schuur et al., 2015). However, several meters of soil can become destabilized within a few days or weeks instead of a few centimeters of soil thawing each year (Schuur et al., 2015). The missing element of the existing studies and models is that the abrupt permafrost destabilization can occur and contribute to more carbon feedback than the existing models predict as the permafrost foundation degrades. The detailed discussion is also given in line 287 to 309 in the revised manuscript.

Reference:

Schuur, E. A., McGuire, A. D., Schädel, C., Grosse, G., Harden, J. W., Hayes, D. J., & Vonk, J. E. (2015). Climate change and the permafrost carbon feedback. Nature, 520(7546), 171-179.

Discussion : In Figure 7c is shown the results of the inversion of shear modulus over the offset distance. The reader can observe a huge value of shear modulus in the permafrost layer located at a offset distance from 500m to 600m. Why this order of magnitude much higher than other parts of the whole profile ? To my mind, this results must be addressed in the discussion as well.

The higher value of shear wave velocity at the Sections 4 and 5 (spanning from 360-600 m, as shown in Figure 4 or Figure 6 in the revised manuscript) is due to the higher value of the R1 wave dispersion curve. As shown in Figure 5b (or Figure B.5 in the revised manuscript), the dispersion curves of the R1 wave at Section 4 and Section 5 are relatively higher than those at the other three sections. The reason for a relatively higher R1 wave velocity in the Sections 4 and 5 could be the

presence of the gravel or larger boulders, as discussed by Glazer et al., 2018 for the testing site. It was reported by Szymański et al. (2013) that this study site also contains a lot of coarse sandy soils and gravels based on the direct sampling methods at the top 15 cm. Figure 4e shows the variation of the shear modulus of soil skeleton predicted by the proposed hybrid inverse and multi-phase poro-mechanical approach. The predicted shear modulus in the first layer at the offset distance of 0 to 360 m ranges from 4 GPa to 7.9 GPa, which represents clay soils (Helgerud et al. 1999). At the offset distance from of 360 to 600 m, the estimated shear modulus in the first layer ranges from 27 GPa to 33 GPa, which corresponds to soils with calcite constituent (Helgerud et al. 1999). Calcite most commonly occurs in sedimentary rock or gravels (Schmid et al., 1987), which is consistent with the field description given by Glazer et al. 2020 and Szymański et al. (2013). The detailed discussion is also given in line 266 to 270 in the revised manuscript.

Reference:

Glazer, M., Dobiński, W., Marciniak, A., Majdański, M., & Błaszczyk, M. (2020). Spatial distribution and controls of permafrost development in non-glacial Arctic catchment over the Holocene, Fuglebekken, SW Spitsbergen. Geomorphology, 358, 107128.

Schmid, S. M., Panozzo, R., & Bauer, S. (1987). Simple shear experiments on calcite rocks: rheology and microfabric. Journal of structural Geology, 9(5-6), 747-778.

Szymański, W., Skiba, S., & Wojtuń, B. (2013). Distribution, genesis, and properties of Arctic soils: a case study from the Fuglebekken catchment, Spitsbergen. Polish Polar Research, 289-304.

[Figure]

Figure 4: Summary of the inversion results at the offset distance from 0 m to 600 m. **(a)** Volumetric ice content distribution. **(b)** Soil porosity distribution. **(c)** Distribution of the degree saturation of unfrozen water. **(d)** Comparison between the numerical and experimental dispersion curves for R2 wave. **(e)** Distribution of the shear modulus of the solid skeletal frame.

[Figure]

Figure 5: Summary of dispersion measurements for Section 1 to 5. **(a)** Dispersion curves of R2 wave. **(b)** Dispersion curves of R1 wave.

L237 : according to this sentence, the ground temperature is deduced from soil temperature among others. How did you get this soil temperature data (modeled, measured on the field ?) ?

The ground temperature was estimated based on an empirical relation. By an empirical relation between the unfrozen water content, porosity, and soil temperature (Liu et al., 2019), we can roughly estimate the average ground temperature distribution in the test site.

The empirical relation is shown in Equation 1.

$$\theta_w = \theta_r + (\theta_{wo} - \theta_r)e^{a(T-T_0)} \tag{1}$$

where $\theta_r$ is the residual volumetric water content; $a$ is the curve fitting parameter; $T_0$ is the freezing temperature considered as $0°C$.

Reference:

Liu, H., Maghoul, P., & Shalaby, A. (2019). Optimum insulation design for buried utilities subject to frost action in cold regions using the Nelder-Mead algorithm. International Journal of Heat and Mass Transfer, 130, 613-639.

Uncertainties : RMS values have to be systematically computed, in order to quantitatively assess the accuracy of all steps of your inversion algorithm. For example, in Figure B3 : why such a misfit between R1 experimental and numerical dispersion curves, comparative to other locations ? I suggest to add a discussion of this issue.

Root mean square value (RMS) has been added in the manuscript to quantify the misfit between experimental and numerical dispersion curves for both R2 and R1 waves (e.g., line 249 and Figure 4 in the revised manuscript).

In our previous inversion analysis, we assumed that the last layer in our model is the unfrozen ground which may be unrealistic considering that the penetrating depth is roughly half of the maximum wavelength (Olafsdottir et al., 2018). For instance, the maximum wavelength in Section 1 is about 22 m (calculated using a phase velocity of 404 m/s at the frequency of 18 Hz). The maximum wavelength for Section 2 to 5 can be calculated in a similar manner. The average maximum wavelength for the entire investigation areas is around 21 m. Therefore, the penetrating depth in the MASW survey presented in this study is only about 11 m. It was reported that the permafrost layer in the studied site can go up to 100 m (Dolnicki et al., 2013; Glazer et al., 2018). Therefore, in the revised paper, we considered the last layer to have a degree of saturation of unfrozen water ranging from 1% to 99%. In this way, the last layer can be either permafrost or unfrozen ground. We have also applied the automatic methods for the selection of dispersion curves (instead of relying on visual inspection that we used in the original draft) using MASWave software (Olafsdottir, 2018). The misfits (RMS) between the R1 experimental and numerical dispersion curves at Section 4 have been significantly reduced from 49.6 to only 11.8, as shown in Figure 6g. The detailed discussion is also given in line 239 to 249 in the revised manuscript.

Reference:

Glazer, M., Dobiński, W., Marciniak, A., Majdański, M., & Błaszczyk, M. (2020). Spatial distribution and controls of permafrost development in non-glacial Arctic catchment over the Holocene, Fuglebekken, SW Spitsbergen. Geomorphology, 358, 107128.

Dolnicki, P., Grabiec, M., Puczko, D., Gawor, L., Budzik, T., & Klementowski, J. (2013). Variability of temperature and thickness of permafrost active layer at coastal sites of Svalbard.

Olafsdottir, E. A., Erlingsson, S., & Bessason, B. (2018). Tool for analysis of multichannel analysis of surface waves (MASW) field data and evaluation of shear wave velocity profiles of soils. Canadian Geotechnical Journal, 55(2), 217-233.

[Figure]

Figure 6: Surface wave inversion results for Section 4 (from 360m to 480m). **(a)** Degree of saturation of unfrozen water, **(b)** Degree of saturation of ice, **(c)** Porosity distribution, **(d)** Shear modulus of solid skeletal frame, **(e)** Bulk modulus of solid skeletal frame, **(f)** Experimental and numerical dispersion curves for R2 wave, **(g)** Experimental and numerical dispersion curves for R1 wave.

**Technical corrections**

Abstract :

L.5 : the term "relatively" is quite imprecise for an abstract, I suggest to remove it.

We have removed the term 'relatively'.

L.7 : "Permafrost and soil layers" is inappropriate, since permafrost are considered as soil as well. Maybe replace it by "active and frozen permafrost layers" ?

We have replaced 'permafrost and soil layers' with 'active and frozen permafrost layers', as shown in line 6 in the revised manuscript.

L.8 : "shear and bulk moduli"

The 'shear modulus and bulk modulus' are widely used in the literature. We think it is still best to keep the current form so that readers can easily follow the definition of those parameters in the three-phase poromechanical model.

Introduction:

L16 to 19 : I would add some references about permafrost thermal definition and permafrost basics.

References have been added for the permafrost definition and its basics, as shown in line 16-19 in the revised manuscript.

Permafrost is defined as the ground that remains at or below 0°C for at least two consecutive years (Riseborough et al., 2008). The shallower layer of the ground in permafrost areas, termed as the active layer, undergoes seasonal freeze-thawing cycles (Shur et al., 2011). The thickness of the active layer depends on local geological and climate conditions such as vegetation, soil composition, air temperature, solar radiation and wind speed (Liu et al., 2019).

Reference:

Riseborough, D., Shiklomanov, N., Etzelmüller, B., Gruber, S., & Marchenko, S. (2008). Recent advances in permafrost modelling. Permafrost and Periglacial Processes, 19(2), 137-156.

Shur Y., Jorgenson M.T., Kanevskiy M.Z. (2011). Permafrost. In: Singh V.P., Singh P., Haritashya U.K. (eds) Encyclopedia of Snow, Ice and Glaciers. Encyclopedia of Earth Sciences Series.

Liu, H., Maghoul, P., Shalaby, A., & Bahari, A. (2019). Thermo-hydro-mechanical modeling of frost heave using the theory of poroelasticity for frost-susceptible soils in double-barrel culvert sites. Transportation Geotechnics, 20, 100251.

L16 : I would replace "upper" by "shallower"

Shallower layer is used, as shown in line 16 in the revised manuscript.

L17 : The expression "freeze-thawing cycles" is more common, maybe replace by it.

We have revised it accordingly, as shown in line 17 in the revised manuscript.

L27 : I would add at least one reference for ice wedge definition.

References to be added for the formation of ice wedge, as shown in line 27 in the revised manuscript.

Ice wedges are large masses of ice formed over many centuries by repeated frost cracking and ice vein growth (Harry et al., 1988).


L64-L65 : I would remove these sentence about potential applications, since you already mention them above. Maybe you can even suggest these application in the discussion and/or conclusion parts.

It has been removed in the revised manuscript.

L70 : remove the article "the" in "for the assessment"

It has been removed in the revised manuscript.

Methods:

L74 : change "the overview" to "an overview"

It has been revised accordingly.

L75 : "surface wave measurements" Maybe you must develop the technique used in details, or precise if these seismic tests are active or passive.

These measurements can be both active and passive since the dispersion model is independent of the seismic source. In this study, the seismic measurements were collected using the active seismic source.

L100-102 : Are this statement and this equation for extracting Rayleigh wave dispersion relation ? If yes, please precise explicitly.

Yes. We have explicitly mentioned this equation is used to describe the dispersion relation of Rayleigh waves, as shown in line 101 in the revised manuscript.

L103 : I would replace "a constant frequency" by "one given frequency"

It has been revised in the manuscript (line 105).

L199 : I would add "respectively" in this sentence

The original sentence in line 199 has been removed in the revised manuscript.

L122 : please, precise what are the two tuning parameters. Are they chosen among the optimization variables mentioned above ?

The two tuning parameters are the number of samples and the number of resampled Voronoi cells (Sambridge 1999), as shown in line 125 in the revised manuscript. These two tuning parameters are different from the optimization variables (physical and mechanical properties).

Sambridge, M. (1999). Geophysical inversion with a neighbourhood algorithm—I. Searching a parameter space. Geophysical journal international, 138(2), 479-494.

L138 : the term "Here" is not clear, you must precise if you mind "in our model" or more focused on one layer of your model.

We have replaced 'here' with 'In this paper', as shown in line 140 in the revised manuscript.

L147 to L159 : This paragraph would be improved by adding some references or figures that illustrates your statements. Actually, it is not very clear for the readers whether the elements are your contribution, or from the current state of the art. For example : the existence of two Rayleigh waves, the respective dependency of R1 and R2 waves to parameters (mechanical and physical). If references exist about these questions, you must add them here. Overall, some physical interpretations will be appreciated : for example, is the higher R1 velocity than R2 velocity easily interpretable in a physical point of view ? Is the difference of sensitivity to physical and mechanical properties between R1 and R2 surprising or expectable ? Why ?

The existence of two Rayleigh waves and the respective dependency of R1 and R2 waves to parameters (physical and mechanical properties) are also the contribution from this research.

To physically interpret the two Rayleigh waves, a uniform frozen soil layer is used to show the propagation of different types of P and S waves and subsequently the formation of Rayleigh waves (R1 and R2) at the surface. It is assumed that an impulse load with a dominant frequency of 100 Hz is applied at the ground surface. The wave propagation analysis was performed in clayey soils by assuming a porosity (n) of 0.5, a degree of saturation of unfrozen water (Sr) of 50%, a bulk modulus (K) of 20.9 GPa and a shear modulus (G) of 6.85 GPa for the solid skeletal frame (helgerud et al., 1999). The velocities of the P1 and P2 waves are calculated as 2,628 m/s and 910 m/s, respectively, based on the relations given in Appendix A in the manuscript. Similarly, the velocities of the S1 and S2 waves are calculated as 1,217 m/s and 481 m/s, respectively. We also found the velocity of R1 and R2 is 1,150 m/s and 450 m/s using the three-phase dispersion relation derived in this paper (Equation 1). It is known that the Rayleigh wave is slightly slower than the shear wave velocity and the ratio of the Rayleigh wave and shear wave velocity ranges from 0.92-0.95 for Poisson's ratio greater than 0.3 (Kazemirad et al., 2013). From this analysis, we found the ratio of R1 and S1 wave velocity is around 0.93. Similarly, the ratio of R2 and S2 wave velocity is around 0.94. Therefore, we can conclude that R1 waves appear due to the interaction of P1 and S1 waves since the phase velocity of R1 waves is slightly slower than the phase velocity of S1 waves. Similarly, R2 waves appear due to the interaction of P2 and S2 waves since the phase velocity of R2 waves is also slightly slower than the phase velocity of S2 waves. This explains why R1 velocity is higher than R2 velocity.

The P1 and S1 waves are strongly related to the longitudinal and transverse waves propagating in the solid skeletal frame, respectively, but are also dependent on the interactions with pore ice and pore water (Carcione and Seriani, 2001). We have proved that R1 waves appear due to the interaction of P1 and S1 waves (see the previous graph). Therefore, the phase velocity of the R1 wave is dependent on both mechanical properties and physical properties. The P2 and S2 waves propagate mainly within pore ice (Leclaire et al., 1994). Hence, the phase velocity of the R2 wave is almost independent of the mechanical properties of the solid skeletal frame, while it is strongly affected by the porosity and degree of saturation of ice.

The detailed discussion is also given in line 141 to 176 in the revised manuscript.


L174 : if you can, precise the type of geophones (type, natural frequency)

The information for geophones has been given in the revised manuscript. The MASW test was performed by using 60 geophone receivers with a frequency of 4.5 Hz spaced at regular 2 m intervals, as shown in line 191-192 in the revised manuscript.

L181 : why "almost completely frozen" ? Precise why you choose the value 85% for the degree of saturation of unfrozen water.

In the revised manuscript to address your comment, we have considered that the degree of saturation of unfrozen water in the active layer is 100% since the test site was extremely wet during the MASW test and the ERT results reported by Glazer et al. (2020) proved that the active layer is most likely completely unfrozen during the MASW testing performed in September. For the permafrost layer (second layer), we have considered that the degree of saturation of unfrozen water is between 1%-85% to be conservative. The degree of saturation of unfrozen water in the third layer is between 1%-100% (permafrost or unfrozen ground, which is to be determined). The detailed discussion is also given in line 204 to 216 in the revised manuscript.

L195 : I would add a reference for illustrating this statement

A reference has been added in line 219-223: 'However, the mechanical properties of permafrost reveal the mineral composition of the soil and soil type (Helgerud et al., 1999), which is valuable in the classification of ice-rich permafrost or even detection of whether the permafrost layer is prone to greenhouse gases carbon dioxide and methane emission to the atmosphere'.

Reference:

Helgerud, M. B., Dvorkin, J., Nur, A., Sakai, A., & Collett, T. (1999). Elastic-wave velocity in marine sediments with gas hydrates: Effective medium modeling. Geophysical Research Letters, 26(13), 2021-2024.

RMS value has been added in line 235-238 in the revised manuscript to quantify the misfits between numerical and experimental dispersion curves: The numerical predictions show good agreement with the experimental dispersion curves for both R1 (RMS value of 1.9) and R2 (RMS value of 4.7) waves.

We have removed original sentence in line 227 in the revised manuscript.

We have removed original sentence in line 239 in the revised manuscript

Discussion and conclusions:

L249 : "makes the analysis more efficient" : you must tell more about this statement : What do you compare this method to ? And, have you done a quantitative assessment to support this discussion?

In this paper, we proposed a separate inversion instead of a joint inversion methods. We firstly used the dispersion of R2 waves to characterize the physical properties of the layers. After obtaining the physical properties. Then the mechanical properties can be derived based on the dispersion relation of the R1 wave mode in a similar manner. The proposed approach (inversion based on R2 and R1 wave modes) in this paper simplifies the inversion of the multi-layered three-phase poromechanical model since the dependent optimization variables are largely reduced. Therefore, the statement of "makes the analysis more efficient" is compared with the case where inversion analysis is performed to determine both physical and mechanical properties at the same time, as shown in line 280 in the revised manuscript.

L255 to 257 : this sentence must be documented by at least one reference.

The original discuss in line 255-257 has been removed in the revised manuscript.

L276 : for the case of a potential early warning system, how do you plan to deal with the seasonal variations (ex: freeze-thawing cycles of the active layer) that you would measure over one year? Do you have any idea how to model and remove such environmental influences that are not related to damage? And how to fix critical values ? If you have any ideas on this issues, you would be welcome to mention them, in order to strengthen your discussion on this potential application.

The applications of our proposed method for early detection and warning systems to monitor infrastructure impacted by permafrost-related geohazards are provided with more details. Active and passive seismic measurements can be collected and processed using the proposed hybrid inverse and poromechanical dispersion model for the assessment and quantitative characterization of permafrost sites at various depths. In the future study, we will focus on the development of an early warning system for the long-term tracking of permafrost conditions. The early warning system can be used to collect seismic measurements and predict the physical and mechanical properties of foundation permafrost in real-time. The system then reports periodic variations of physical (mostly ice content) and mechanical properties of the permafrost being monitored. The same method being applied on different times (e.g. seasonal basis) can be used to record the change of properties of the permafrost site, and then warn on the level of degradation of the permafrost exceeding the threshold. The value of the threshold (or critical values) will require more in-depth research to be determined. The early detection and warning systems can be beneficial in monitoring the state of the foundation permafrost and preventing excessive thawing settlement and significant loss in strength. Similarly, we can detect the presence of peat (based on the physical and mechanical properties) which is vulnerable to permafrost carbon feedback and emission of greenhouse gases into the atmosphere. It's reported the soils in the permafrost region hold twice as much carbon as the atmosphere does (almost 1,600 billion tonnes) (Schuur et al., 2015). The thawing permafrost can rapidly trigger landslides and erosion. Current climate models assume that permafrost thaws gradually from the surface downwards (Schuur et al., 2015). However, several meters of soil can become destabilized within a few days or weeks instead of a few centimeters of soil thawing each year (Schuur et al., 2015). The missing element of the existing studies

and models is that the abrupt permafrost destabilization can occur and contribute to more carbon feedback than the existing models predict as the permafrost foundation collapses. The detailed discussion is also given in line 287 to 309 in the revised manuscript.


Appendices:

References (Carcione & Seriani (2001) and Leclaire (1994)) have been added in Appendices A, C and D in the revised manuscript.


Appendix B : in all figures you show both saturation degree of unfrozen water and saturation degree of ice, but only one seems to be useful, since the two variables are directly linked together. Furthermore, what about the results of the layer thickness from this surface wave inversion? It could be appropriate to show them as well. Again, for R1 and R2 experimental and numerical dispersion curves, it should be good to precise misfits through RMS values.

Even though the degree of saturation of unfrozen water is directly related to the degree of saturation of ice, we still think it might be helpful to include both of them. The plot of the degree of saturation of unfrozen water can directly show readers the characteristic of the active layer; whereas the degree of saturation of ice can directly show readers the characteristic of permafrost layers. Also, the thickness is given in the vertical axis. We also added the RMS values for the comparison between the experimental and numerical dispersion curves for both R1 and R2 waves in the revised manuscript.

L297 : I suggest to add "respectively"

We have added 'respectively' in this sentence, as shown in line 330 in the revised manuscript.

L382 : "Convention" instead of "convection"

It has been corrected in line 416 in the revised manuscript.

L396 : "The values of each component" instead of "The value of each components"

It has been corrected in line 425 in the revised manuscript.

Appendix D : L432 : I suggest to remove "the matrix formed by" for consistency

It has been removed in the revised manuscript.

**Appendix A: Definition of Phase Velocities**

The velocities of the three types of P waves are determined by a third degree characteristic equation (Leclaire et al., 1994 and Carcione et al., 2000):

$$\Lambda^3 \tilde{R} - \Lambda^2 \left( (\rho_{11}\tilde{R}_{iw} + \rho_{22}\tilde{R}_{si} + \rho_{33}\tilde{R}_{sw}) - 2(R_{11}R_{33}\rho_{23} + R_{33}R_{12}\rho_{12}) \right)$$
$$+ \Lambda \left( (R_{11}\tilde{\rho}_{iw} + R_{22}\tilde{\rho}_{si} + R_{33}\tilde{\rho}_{sw}) - 2(\rho_{11}\rho_{23}R_{23} + \rho_{33}\rho_{12}R_{12}) \right) - \tilde{\rho} = 0$$

where

$$\tilde{R} = R_{11}R_{22}R_{33} - R_{23}^2 R_{11} - R_{12}^2 R_{33}$$
$$\tilde{R}_{sw} = R_{11}R_{22} - R_{12}^2$$
$$\tilde{R}_{iw} = R_{22}R_{33} - R_{23}^2$$
$$\tilde{R}_{si} = R_{11}R_{33}$$
$$\tilde{\rho} = \rho_{11}\rho_{22}\rho_{33} - \rho_{23}^2 \rho_{11} - \rho_{12}^2 \rho_{33}$$
$$\tilde{\rho}_{sw} = \rho_{11}\rho_{22} - \rho_{12}^2$$
$$\tilde{\rho}_{iw} = \rho_{22}\rho_{33} - \rho_{23}^2$$
$$\tilde{\rho}_{si} = \rho_{11}\rho_{33}$$

The roots of the third degree characteristic equation, denoted as $\Lambda_1$, $\Lambda_2$ and $\Lambda_3$, can be found by computing the eigenvalues of the companion matrix (Horn and Johnson, 2012). The velocities of the three types of P-wave ($v_{p1} > v_{p2} > v_{p3}$) are given as follows:

$$v_{p1} = \sqrt{\frac{1}{\Lambda_1}}; \quad v_{p2} = \sqrt{\frac{1}{\Lambda_2}}; v_{p3} = \sqrt{\frac{1}{\Lambda_3}}$$

The velocities of the two types of S-wave are determined by a second degree characteristic equation:

$$\delta^2 \rho_{22}\tilde{\mu}_{si} - \delta(\mu_{11}\tilde{\rho}_{iw} + \mu_{33}\tilde{\rho}_{sw}) + \tilde{\rho} = 0$$

The roots of this second degree characteristic equation is denoted by $\delta_1$ and $\delta_2$. The velocities of the two types of S-wave ($v_{s1} > v_{s2}$) are given as follows:

$$v_{s1} = \sqrt{\frac{1}{\delta_1}}; \quad v_{s2} = \sqrt{\frac{1}{\delta_2}};$$

Reviewer 2

**General Comments**

The main contribution of this article is the identification of two wave modes that interact sensitively with the porous and mechanical properties of a layered poro-elastic medium, respectively. In addition the authors use a multiphase poro-mechanical foundation to build a global matrix model of a layered poro-elastic medium. This approach has significant novelty compared to the typical global matrix models based on partial wave amplitudes and nicely emphasizes the connection between surface waves and ground physical properties. The specific application to permafrost demonstrates the relevance of this manuscript to The Cryosphere, although certain aspects of the methodology also have broader relevance to the field of near surface geophysics and non-destructive testing of engineering materials.

The weaker side of the manuscript is that findings that are a direct result of the data examples presented are not adequately separated from other applications that remain essentially hypothetical (these are detailed under "specific comments").

In this paper, our results demonstrate the potential of seismic surface wave testing accompanied with our proposed hybrid inverse and poromechanical dispersion model for the assessment and quantitative characterization of permafrost sites, as indicated in line 70-72 in the revised manuscript. We have clarified that its applications for early detection and warning systems to monitor infrastructure impacted by permafrost-related geohazards, and to detect the presence of layers vulnerable to permafrost carbon feedback and emission of greenhouse gases into the atmosphere will be the goal of our future studies. The detailed discussion is given in line 285-307 in the revised manuscript.

The manuscript could also be improved substantially by giving a more complete description of the two Rayleigh wave modes so that those reading the manuscript may better understand their propagation and interpret their broader relevance to the field of surface wave seismic investigation.

More information of the two Rayleigh wave modes (R1 and R2) is given to better explain their propagation in permafrost foundations (details are given in the answer of Question 15 as well as line 148-162 in the revised manuscript).

Furthermore, I have some concerns that the inversion results are overly sensitive to the frequency range of the dispersion curves that constrain the inversion (likely due to a mismatch between the shape of the experimental and inverted dispersion curves). The anomalous result at 360-480 m in the physical data example is not convincing and appears more likely to reflect a weakness in the inversion methodology than real lateral variation in physical properties.

In our inversion analysis in the original manuscript, we assumed that the last layer in our model is the unfrozen ground, which is indeed uncertain considering that the penetrating depth is only about 11 m in the MASW survey (based on the recommendation that MASW investigation depth is roughly half of the maximum wavelength (Olafsdottir et al., 2018)). For instance, the maximum wavelength in Section 1 is about 22 m (calculated using a phase velocity of 404 m/s at the frequency of 18 Hz). The maximum wavelength for Section 2 to 5 can be calculated in a similar manner. The average maximum wavelength for the entire investigation areas is around 21 m. Therefore, the penetrating depth in the MASW survey presented in this study is only about 11 m.

It was reported that the permafrost layer in the studied site can go up to 100 m (Dolnicki et al., 2013; Glazer et al., 2018). Therefore, in the revised paper, we considered the last layer to have a degree of saturation of unfrozen water ranging from 1% to 99%. In this way, the last layer can be either permafrost or unfrozen ground. We have also applied the automatic methods for the selection of dispersion curves (instead of relying on visual inspection that we used in the original draft) using MASWave software (Olafsdottir, 2018). The misfits (RMS) between the R1 experimental and numerical dispersion curves at Section 4 have been significantly reduced from 49.6 to only 11.8 , as shown in Figure 6g (or Figure B.3 in the revised manuscript). The detailed discussion is also given in line 193-203 in the revised manuscript.

Furthermore, it is generally difficult to assess the true accuracy of the results owing to a lack of comparison to ground truth observations of physical properties and interface depths or comparison with other geophysical datasets (both of which appear to exist in the published literature). I believe it should be possible for the authors to address these concerns in a revised manuscript, that will then make a useful contribution to The Cryosphere.

In the revised manuscript, the comparison of the inversion results using the proposed hybrid inverse and multi-phase poro-mechanical approach and inversion results from ERT survey provided by Glazer et al., (2020) has been added in line 251-268. It was reported by Glazer et al., (2020) that the permafrost table is located at a depth of about 2 m for a span of 20 m. The new inversion results in terms of the thickness of the active layer were also validated using the results reported by Dobiński et al., (2010) and Dolnicki et al., (2013) by the direct probing method. It was also reported by Dobiński et al., (2010) and Dolnicki et al., (2013) that the active layer in Svalbard is approximately 1.65–2.5 m deep. The direct sampling results reported by Szymański et al. (2013) confirmed that the study site is very wet and the water table is very high (around 15 cm). It was reported by Szymański et al. (2013) that this study site also contains a lot of coarse sandy soils, gravels as well as around 20% silty clay based on the direct sampling methods at the top 15 cm. Our inversion results, as shown in Figure 4, predicted that the permafrost table is generally located at about 1.5-1.9 m below the ground surface, which is consistent with the ERT results reported by Glazer et al., (2020) and results reported by Dobiński et al., (2010) and Dolnicki et al., (2013) using the direct probing method.

Based on the field description of the testing site by Glazer et al., (2020), the unconsolidated sedimentary rock contains a high proportion of pore spaces; consequently, they can accumulate a large volume of pore-water or pore-ice. Our inversion results showed that the porosity of the active layer ranges from 0.56 to 0.69, which is consistent with the field description by Glazer et al., (2020). The unfrozen water content in the second permafrost layer was predicted ranging from 0.05-0.17. Li et al. (2020) and Zhang et al. (2020) showed that the residual volumetric unfrozen water content for silty-clay, clay, medium sand, and fine sand is 0.12, 0.08, 0.06 and 0.03, respectively. Our inversion results predicted that soils are mostly silty-clay or clay (Section 1-3) and sandy soils, which are also consistent with the results described by Szymański et al. (2013). Figure 4e shows the variation of the shear modulus of soil skeleton predicted by the proposed hybrid inverse and multi-phase poro-mechanical approach. The predicted shear modulus in the first layer at the offset distance of 0 to 360 m ranges from 4 GPa to 7.9 GPa, which represents clay soils (Helgerud ET AL. 1999). At the offset distance of 360 to 600 m, the estimated shear modulus in the first layer ranges from 27 GPa to 33 GPa, which corresponds to soils with calcite constituents (Helgerud ET AL. 1999). Calcite most commonly occurs in sedimentary rock or gravels (Schmid et al., 1987), which is consistent with the field description given by Glazer et al. 2020 and Szymański et al.

(2013).

The above discussion is also given in line 239-270 in the revised manuscript.

"It is usually assumed that Rayleigh waves are the prevailing type of waves generated, with a depth penetration of about one wavelength (Viktorov 1967). However, it has been reported that this assumption holds strictly only at sites where the stiffness increases smoothly as a function of depth (Foti 2000). At sites with a velocity reversal (i.e. stiffness decreases with depth), the nature of surface-wave propagation has been reported as more complicated than at sites with normal dispersion. Several studies have indicated that a measured dispersion curve where the phase velocity increases with frequency, i.e. inverse dispersion, is actually built up by small portions of higher modes (Gucunski and Woods 1992; Tokimatzu et al. 1992; Forbriger 2003; Foti et al. 2003; Ryden et al. 2004)."

The Rayleigh-Lamb waves are formed by interference of multiple reflections and mode conversion of compressional waves (P-waves) and shear waves (S-waves) at the two free boundaries of the plate. Strictly speaking, Lamb's theory for these guided waves requires the surfaces of the plate to be traction-free, that is to say the plate should be in a vacuum (Lowe 2001). For pavement

structure studied by Ryden & Park (2004), they concluded: 'Lamb-wave dispersion curves for a free plate in a vacuum can represent the reality sufficiently closely only if the stiffness contrast between the top layer and the underlying half-space is large. The resulting error in phase velocity in this case was investigated. It was concluded that the error in phase velocity does not exceed 5% if the fundamental antisymmetric Lamb-wave dispersion curve is used as an approximate theoretical dispersion curve, with the restriction that the shear-wave velocity of the stiff layer is greater than the compressional-wave velocity in the underlying media". For the wave propagation in permafrost, the condition that the shear-wave velocity of the stiff layer (first layer) is greater than the compressional-wave velocity in the underlying media can not be fulfilled.

As mentioned earlier, the formation of guided waves requires multiple reflections and mode conversion of compressional waves (P-waves) and shear waves (S-waves). In the original seismic measurements shown in Figure 2e, we can not see any reflection that can be easily seen in any traditional seismic reflection investigations (e.g., French 1975; Zhao et al., 1993; Symes 2009).

Additionally, the phase velocity of R1 and R2 does not converge to the same velocity as the Rayleigh-Lamb waves do since the generation of R1 or R2 as well as symmetric modes or anti-symmetric modes are very different. The fundamental anti-symmetric mode (A0) and symmetric mode (S0) of Rayleigh-Lamb waves tend to converge to the same phase velocity (as described by Ryden & Park (2004)). The symmetric modes, also called longitudinal modes, are generated due to the wave propagation in the longitudinal direction. On the other hand, the antisymmetric modes are generated because of the wave propagation in the transverse direction (Graff 2012). The generation of R1 and R2 is due to the interaction of multiphase components (soil skeleton, unfrozen water and ice), as discussed in line 148-161 in the revised manuscript. We have proved that R1 waves appear due to the interaction of P1 and S1 waves. Similarly, R2 waves appear due to the interaction of P2 and S2 waves. However, the generation of anti-symmetric mode and symmetric mode is due to the geometry that constrains the wave propagation. The anti-symmetric mode and symmetric mode can be seen in solid materials. However, the R1 and R2 can only be generated in frozen soils with at least three phases.

For the dispersion curve of R1 mode, it is very different from the dispersion curve build up from the higher mode. If the dispersion curve is built up by small portions of higher modes, we can easily see the cutting off frequency (as shown by Romeyn et al., 2021 in Figure 7 in their manuscript). This is not the case for the dispersion curve shown in Figure 2f (or Figure 3 in the revised manuscript) in our study.


To further explain this statement, we have shown a dispersion image obtained from the three-phase poro-mechanical approach for a three-layer system. We assumed that the porosity is 0.5 for all three layers; the degree of saturation of unfrozen water is 0.1, 0.3 and 0.6, respectively; the shear modulus of soil skeleton is 6.85 GPa, 10 GPa and 10 GPa, respectively; the bulk modulus of soil skeleton is 15 GPa, 15 GPa and 21 GPa, respectively. The image contains two colors (red and blue). The interface of two colors indicates the sign switching of determinant value, which is the definition of dispersion relation. Figure 7a (or Figure E.1 in the revised manuscript) shows the dispersion image (a combination for R1 and R2 waves) calculated using the proposed three-phase poro-mechanical approach. Figure 7b shows the dispersion image using the components related only to the P1 and S1 wave velocities. Figure 7c shows the dispersion image using the components related only to the P2 and S2 wave velocities. Therefore, we can conclude that the global stiffness matrix for the R1 wave can be decomposed into the components related only to the P1 and S1 wave velocities. This approach avoids the difficulties in differentiating the higher modes of R2 wave from the fundamental mode of the R1 wave. The detailed discussion is also given in Appendix E in the revised manuscript.

[Figure]

Figure 7: **(a)** Dispersion image (a combination for R1 and R2 waves) **(b)** Dispersion image using the components related only to the P1 and S1 wave velocities. **(c)** Dispersion image using the components related only to the P2 and S2 wave velocities.


frozen state of the permafrost. However, the frozen permafrost has elevated porosity compared to both the overlying and underlying layers. If we consider that water plus ice saturation in the permafrost is at least 86% of a frozen pore volume of 0.43-0.46, then it will likely be more than 100% saturation if the porosity in the unfrozen state is approximated by the porosity of the overlying active layer and underlying unfrozen ground (represented as 0.34-0.36 if we take the zone of overlap between the two porosity estimates). I think the authors should explicitly step through their assumptions and calculations in determining the stability of the permafrost, because this is a significant application area of their methodology and could be a major strength of the work if developed to its full potential.

Authors agree that unstable permafrost is distinguished by whether the ice and water saturation exceed the total pore volume of the ground in an unfrozen state. However, in our case it is almost impossible to accurately evaluate the total pore volume of permafrost in its unfrozen state since we would have to conduct the seismic test when the permafrost thaws.

Here, we consider the segregated ice plays the same role as pore-ice from a continuum mechanics point of view. The growth of ice lenses is approximated as an increase in the soil porosity in a Representative Elementary Volume (REV) (the porosity is not the same as the unfrozen state since we have volumetric expansion in freezing process. Instead, as Michalowski et al. 2006 described, the porosity increases with the growth of ice lenses. Therefore, the total pore volume saturated with pore ice and unfrozen water is always the same as the porosity). Therefore, the determined volumetric ice content (Figure 4) can correspond to both pore-ice and segregated ice (ice lenses) as an average value. The degree of saturation of ice in permafrost can be roughly used to indicate whether permafrost is stable or not. In our revised inversion results, it is predicted that we have a low degree of saturation of unfrozen water (or a high degree of saturation of ice) , as shown in Figure 4c. The relatively higher degree of saturation of unfrozen water at the offset distance from 120 m to 360 m can be due to the seawater infiltration. Due to freezing point depression contributed by the seawater infiltration (Wu et al. 2017), unfrozen water content at the offset distance from 120 m to 360 m is expected to be considerably higher than that at other offsets. However, these discussions have been removed from the revised manuscript.


Figure 3a shows the probabilistic distribution of the degree of saturation of unfrozen water with depth in Section 1. Our results show that the active layer has a thickness of about 1.5 m. The predicted permafrost layer (second layer) has a nearly 32% of degree of saturation of unfrozen pore water. Figure 3b shows the degree of saturation of ice with depth. The degree of saturation of ice in the permafrost layer (second layer) ranges from 67% to 79%. Figure 3c illustrates the porosity distribution with depth. The porosity is around 0.60 in the first layer (active layer), from 0.40 to 0.47 in the second layer (permafrost) and from 0.56 to 0.59 in the third layer. Figure 3d and 3e show the predicted mechanical properties of the solid skeletal frame (shear modulus and bulk modulus) in each layer. It was reported by Szymański et al. (2013) that this study site also contains a lot of coarse sandy soils, gravels as well as around 20% silty clay based on the direct sampling methods at the top 15 cm. The predicted shear modulus and bulk modulus for the solid skeletal frame in the permafrost layer (second layer) are about 13 GPa and 12.7 GPa, which are in the range for silty-clayey soils (Vanorio et al. 2003) and are also consistent with the local soil types described by Szymański et al. (2013). The predicted shear modulus and bulk modulus for the solid skeletal frame in the third layer are about 4 GPa and 10 GPa, which are in the range for clayey soils (Vanorio et al. 2003). Figure 3f and 3g show the comparison between the numerical and experimental dispersion relations for R2 and R1 waves, respectively. The numerical predictions show good agreement with the experimental dispersion curves for both R1 and R2 waves. The uncertainty analyses for other Sections are performed in a similar manner. The detailed discussion is also given in line 224-238 in the revised manuscript.


We have replaced it with 'degree of saturation' in those figures in the revised manuscript.

Reviewer 3

**General Comments**

This paper reports on the inversion of the dispersion curves from two types of seismic waves propagating in permafrost, to evaluate soil properties via a three-phase analytical model that accounts for the porosity and mechanical properties in the three layers. I think this paper is well written and is of interest for the audience of the Cryosphere. The findings are convincing, and in my opinion the paper deserves publication after the comments listed below have been considered.

For a non-specialist of permafrost-related studies, it would be beneficial that the authors provide a more systematic comparison of their approach with already existing methods. The main novelty here is the fact that the forward model includes new parameters (such as porosity, and the degree of ice saturation), and that the two waves are sensitive to different sets of parameters, which allows a separate inversion instead of a joint inversion. This should be emphasized by citing previous investigations of permafrost with seismic methods, and by checking that the parameters are consistent with other similar studies, when possible.

Currently, the existing models predominately apply an elastic approach as the forward solver in the dispersion analysis for permafrost sites. Our proposed model, on the other hand, uses a hybrid inverse and multi-phase poroelastic approach for the characterization of permafrost sites. In the existing methods that are used for the inversion analysis of MASW measurements for permafrost sites, it is commonly considered that the permafrost layer (frozen soil) is associated with a higher shear wave velocity due to the presence of ice in comparison to the unfrozen ground (Dou et al, 2014, Glazer et al, 2020). However, the porosity and soil type can also significantly affect the shear wave velocity (Liu et al, 2020). In other words, a relatively higher shear wave velocity could be associated with an unfrozen soil layer with a relatively lower porosity or stiffer solid skeletal frame, and not necessarily related to the presence of a frozen soil layer. Therefore, the detection of the permafrost layer from only the shear wave velocity may lead to inaccurate and even misleading interpretations. The detailed discussion is also given in line 52-61 in the revised manuscript.

Here, we present a hybrid inverse and multi-phase poromechanical approach for physics-based in-situ characterization of permafrost sites using surface wave techniques. In our method, we quantify the physical properties such as ice content, unfrozen water content, and porosity as well as the mechanical properties such as the shear modulus and bulk modulus of permafrost or other soil layers. The amount of ice content can be used, rather than the shear wave velocity, to explicitly indicate the active layer, permafrost and unfrozen ground. The role of two different types of Rayleigh waves in characterizing the permafrost is presented based on an MASW seismic investigation in a field located at SW Spitsbergen, Svalbard. As mentioned in your comment, the objective is to use a separate inversion instead of a joint inversion. Glazer et al, 2020 performed both seismic surveys (MASW test) and electrical resistivity investigations at the site in September 2017 to study the evolution and formation of permafrost considering surface watercourses and marine terrace. In our study, the same experimental data collected by Glazer et al, 2020 is used to demonstrate the inversion analysis based on R1 and R2 Rayleigh waves. The detailed discussion is also given in line 62-72 in the revised manuscript.

In the revised manuscript, the comparison of the inversion results using the proposed hybrid inverse and multi-phase poro-mechanical approach and inversion results from the ERT survey provided by Glazer et al., (2020) has been added. It was reported by Glazer et al., (2020) that the permafrost table is located at a depth of about 2 m for a span of 20 m investigated by the ERT

survey. The new inversion results in terms of the thickness of the active layer were also validated using the results reported by Dobiński et al., (2010) and Dolnicki et al., (2013) by the direct probing method. It was also reported by Dobiński et al., (2010) and Dolnicki et al., (2013) that the active layer in Svalbard is approximately 1.65–2.5 m deep. The direct sampling results reported by Szymański et al. (2013) confirmed that the study site is very wet and the water table is very high (around 15 cm). It was reported by Szymański et al. (2013) that this study site also contains a lot of coarse sandy soils, gravels as well as around 20% silty clay based on the direct sampling methods at the top 15 cm. Our inversion results, as shown in Figure 4, predicted that the permafrost table is generally located at about 1.5-1.9 m below the ground surface, which is consistent with the ERT results reported by Glazer et al., (2020) and results reported by Dobiński et al., (2010) and Dolnicki et al., (2013) using the direct probing method.

Based on the field description of the testing site by Glazer et al., (2020), the unconsolidated sedimentary rock contains a high proportion of pore spaces; consequently, they can accumulate a large volume of pore-water or pore-ice. Our inversion results showed that the porosity of the active layer ranges from 0.56 to 0.69, which is consistent with the field description by Glazer et al., (2020). The unfrozen water content in the second permafrost layer was predicted ranging from 0.05-0.17. Li et al. (2020) and Zhang et al. (2020) showed that the residual volumetric unfrozen water content for silty-clay, clay, medium sand, and fine sand is 0.12, 0.08, 0.06 and 0.03, respectively. Our inversion results predicted that soils are mostly silty-clay or clay (Section 1-3) and sandy soils, which are also consistent with the results described by Szymański et al. (2013). Figure 4e shows the variation of the shear modulus of soil skeleton predicted by the proposed hybrid inverse and multi-phase poro-mechanical approach. The predicted shear modulus in the first layer at the offset distance of 0 to 360 m ranges from 4 GPa to 7.9 GPa, which represents clay soils (Helgerud et al. 1999). At the offset distance of 360 to 600 m, the estimated shear modulus in the first layer ranges from 27 GPa to 33 GPa, which corresponds to soils with calcite constituents (Helgerud et al. 1999). Calcite most commonly occurs in sedimentary rock or gravels (Schmid et al., 1987), which is consistent with the field description given by Glazer et al. 2020 and Szymański et al. (2013). The above discussion is also given in line 253-270 in the revised manuscript.


Figure 4. I am not sure that the terminology employed to describe these two waves is adequate. The spectra look similar to those encountered when dealing with a mix between surface and guided waves, which are also created by the interference between P and S waves. Leaky guided waves are encountered in configurations where the layers have impedance discontinuities, for

example sea ice on water, seismic waves in roads (a hard layer of bitumen resting on a soft substrate). Moreover, the R2 wave seems to have a cutoff frequency around 12 Hz, which indicates a higher-order mode. This needs clarifying. This could also be another explanation for the higher misfit in figure 7d. Have you checked the polarization of the two waves?

The guided waves are formed by interference of multiple reflections and mode conversion of compressional waves (P-waves) and shear waves (S-waves) at the two boundaries of the plate. The fundamental anti-symmetric mode (A0) and symmetric mode (S0) tend to converge to the same phase velocity (as described by Ryden & Park (2004)). The symmetric modes, also called longitudinal modes, are generated due to the wave propagation in the longitudinal direction. On the other hand, the antisymmetric modes are generated because of the wave propagation in the transverse direction (Graff 2012). In our study, the phase velocity of R1 and R2 does not converge to the same velocity since the generation of R1 or R2 as well as symmetric modes or antisymmetric modes are fundamentally different. The generation of R1 and R2 is due to the interaction of multiphase components (soil skeleton, unfrozen water and ice). We have proved that R1 waves appear due to the interaction of P1 and S1 waves. Similarly, R2 waves appear due to the interaction of P2 and S2 waves. However, the generation of anti-symmetric mode and symmetric mode is due to the geometry that constrains the wave propagation. The anti-symmetric mode and symmetric mode can be seen in solid materials. However, the R1 and R2 can only be generated in frozen soils with at least three phases.

The R2 wave at a range of 18-33 Hz is extracted from the original signal. Since the signal does not have clear low-frequency component, the exact cutting off frequency of R2 can not be determined. This has been discussed by Park et al. 2007. In their publication, dispersion curve shown in Figure 6 also shows a similar dispersion spectra where only dispersion curve after 18 Hz can be clearly identified (Park et al. 2007).

In our previous inversion analysis, we assumed that the last layer in our model is the unfrozen ground, which is indeed uncertain considering that the penetrating depth is only about 11 m in the MASW survey (based on the recommendation that MASW investigation depth is roughly half of the maximum wavelength (Olafsdottir et al., 2018)). For instance, the maximum wavelength in Section 1 is about 22 m (calculated using a phase velocity of 404 m/s at the frequency of 18 Hz). The maximum wavelength for Section 2 to 5 can be calculated in a similar manner. The average maximum wavelength for the entire investigation areas is around 21 m. Therefore, the penetrating depth in the MASW survey presented in this study is only about 11 m. It was reported that the permafrost layer in the studied site can go up to 100 m (Dolnicki et al., 2013; Glazer et al., 2018). Therefore, in the revised paper, we considered the last layer to have a degree of saturation of unfrozen water ranging from 1% to 99%. In this way, the last layer can be either permafrost or unfrozen ground. We have also applied the automatic methods for the selection of dispersion curves (instead of relying on visual inspection that we used in the original draft) using MASWave software (Olafsdottir, 2018). The misfits (RMS) between the R1 experimental and numerical dispersion curves at Section 4 have been significantly reduced from 49.6 to only 11.8 , as shown in Figure 6g (or Figure B.3g in the revised manuscript).


Figure 3a shows the probabilistic distribution of the degree of saturation of unfrozen water with depth in Section 1. Our results show that the active layer has a thickness of about 1.5 m. The predicted permafrost layer (second layer) has a nearly 32% of degree of saturation of unfrozen pore water. Figure 3b shows the degree of saturation of ice with depth. The degree of saturation of ice in the permafrost layer (second layer) ranges from 67% to 79%. Figure 3c illustrates the porosity distribution with depth. The porosity is around 0.60 in the first layer (active layer), from 0.40 to 0.47 in the second layer (permafrost) and from 0.56 to 0.59 in the third layer. Figure 3d and 3e show the predicted mechanical properties of the solid skeletal frame (shear modulus and bulk modulus) in each layer. It was reported by Szymański et al. (2013) that this study site also contains a lot of coarse sandy soils, gravels as well as around 20% silty clay based on the direct sampling methods at the top 15 cm. The predicted shear modulus and bulk modulus for the solid skeletal frame in the permafrost layer (second layer) are about 13 GPa and 12.7 GPa, which are in the range for silty-clayey soils (Vanorio et al. 2003) and are also consistent with the local soil types described by Szymański et al. (2013). The predicted shear modulus and bulk modulus for the solid skeletal frame in the third layer are about 4 GPa and 10 GPa, which are in the range for clayey soils (Vanorio et al. 2003). Figure 3f and 3g show the comparison between the numerical and experimental dispersion relations for R2 and R1 waves, respectively. The numerical predictions show good agreement with the experimental dispersion curves for both R1 and R2 waves. The uncertainty analyses for other Sections are performed in a similar manner. The detailed discussion is also given in line 224-238 in the revised manuscript.

---

## Author Response (AR2)

**Subject:** Detailed Responses to Reviewers: Liu, H., Maghoul, P., and Shalaby, A.: Seismic physics-based characterization of permafrost sites using surface waves, The Cryosphere Discuss. [preprint], `https://doi.org/10.5194/tc-2021-219`, in review, 2021.

**Date:** February 22, 2022
* * *
The authors are grateful for the valuable comments and kind consideration of our submission. Detailed responses and revisions based on these comments are listed below.

The third layer of your model needs more descriptions, perhaps, with some modifications to figure labels (Fig 4 and B.1-B.4). This should not take much effort.

We have provided more description for the third layer in line 210-213 in the revised manuscript. The degree of saturation of unfrozen water in the third layer is between 1%-100% (permafrost or unfrozen ground, which is to be determined). In our analysis, the third layer is assumed to be infinite. However, with the limited investigation depth constrained by the wavelength of the performed MASW tests, the inversion results beyond the maximum investigation depth are not considered in the paper.

It would be much appreciated if reviewers can specify the suggested modifications to the figure labels.

Line 18 (Shur Y., 2011) to (Shur, 2011)

We have corrected the reference Shur et al., 2011 in line 18 in the revised manuscript.

Line 159: Figure 2c and 2d illustrate into Figures.

We have corrected the 'Figure' into 'Figures' in line 159 in the revised manuscript.

Lines 178, 180 by (Glazer et al., 2020) to by Glazer et al. (2020).

Based on the recommended citation style by The Cryosphere, the origical citation style (Glazer et al., 2020) seems appropriate.

Line 182 has a a thick - please, remove redundant "a".

It has been removed in line 182 in the revised manuscript.

Lines 185, 234 by (Szymanski et al., 2013) to by Szymanski et al. (2013) .

Based on the recommended citation style by The Cryosphere, the origical citation style seems appropriate.

Line 193 Figure 3b, 3c ... show to Figures 3b, 3c ... .

We have corrected the 'Figure' into 'Figures' in line 193 in the revised manuscript.

We have corrected the 'Figure' into 'Figures' in line 195 and 203 in the revised manuscript.

We have added space in Fig. 4 caption.

Fig. 6: I could not follow the figure caption. Please check these two entries: "(c) Distribution of the shear modulus of the solid skeletal frame. "- but (c) shows the "Degree of saturation of unfrozen water"; while (e) provides the Shear modulus. "(e) Predicted average soil temperature distribution."- but (e) shows the shear modulus, while there is no subplot about the temperature..

We have updated the caption for Fig .6. (c): Degree of saturation of unfrozen water distribution. (e): Distribution of the shear modulus of the solid skeleton.

Lines 262, 479: "poro-mechanical" in the rest of the paper poromechanical has been used, please choose your favorite. .

We have replaced 'poro-mechanical' with 'poromechanical'.

We have updated them in line 341 and 350 in the revised manuscript.

We have updated them in line 412 in the revised manuscript.

We have updated it in line 448 in the revised manuscript.

Line 477 Figure E.1 contains two colors (red and blue) - there is no red (green and blue).

We have updated it (green and grey) in line 480 in the revised manuscript.

We have updated it in Fig. E1 caption in the revised manuscript.

Appendix F: In several equations, I could not follow the difference between $p_{33}$ and $P_{33}$.

$P_{33}$ should be $p_{33}$ and we have corrected it in Appendix F.

References: Albaric et al., 2021 - the correct journal name should be Seismol. Res. Lett. Dolnicki et al., 2013 - bibliographic information is incomplete.

We have updated references in the revised manuscript.